# Accurate surrogate amplitudes with calibrated uncertainties

Henning Bahl[1], Nina Elmer[1], Luigi Favaro[1,2], Manuel Haußmann[3],
Tilman Plehn[1,4] and Ramon Winterhalder[5]

**1** Institut für Theoretische Physik, Universität Heidelberg, Germany
**2** CP3, Université catholique de Louvain, Louvain-la-Neuve, Belgium
**3** Department of Mathematics and Computer Science, University of Southern Denmark
**4** Interdisciplinary Center for Scientific Computing (IWR), Universität Heidelberg, Germany
**5** TIFLab, Universitá degli Studi di Milano & INFN Sezione di Milano, Italy

## Abstract

**Neural networks for LHC physics have to be accurate, reliable, and controlled. Using neural surrogates for the prediction of loop amplitudes as a use case, we first show how activation functions are systematically tested with Kolmogorov-Arnold Networks. Then, we train neural surrogates to simultaneously predict the target amplitude and an uncertainty for the prediction. We disentangle systematic uncertainties, learned by a well-defined likelihood loss, from statistical uncertainties, which require the introduction of Bayesian neural networks or repulsive ensembles. We test the coverage of the learned uncertainties using pull distributions to quantify the calibration of cutting-edge neural surrogates.**

| | |
|---|---|
| Received | 2024-12-19 |
| Accepted | 2025-09-25 |
| Published | 2025-10-24 |

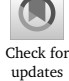

# 1  Introduction

The goal of particle physics is to identify the fundamental properties of particles and their interactions, going beyond the current Standard Model. It is based on an optimal interplay of precision predictions and precision measurements. This precision requirement is constantly challenged by the vast amount of recorded and simulated data, the complexity of the data, and the need to use all aspects of the data for an optimal analysis.

The revolution of LHC physics through modern machine learning (ML) is happening right now [1, 2]. On the theory and simulation side, we can use neural networks to improve every aspect of our simulation chain, starting from phase-space sampling [3–11] to scattering amplitude evaluations [12–22], end-to-end event generation [23–27], and ultra-fast detector simulations [28–49].

The workhorses for this transformation are surrogate and generative networks, which have been shown to reliably interpolate training data [50, 51]. Given the LHC requirements, they have to be controlled and precise in encoding kinematic patterns over an essentially interpretable phase space [52–56]. Conditional versions of generative networks enable new analysis methods, like probabilistic unfolding [57–64] or inference through posterior sampling [65–67].

Going back to LHC event generation — once the ML-simulation tools are fast enough to provide sufficient numbers of simulated events, the challenge in theory predictions shifts to fast implementation of higher-order perturbation theory. Here we need to ensure that network surrogates trained on theory predictions are controlled and precise enough so they can reflect the underlying theory precision. Surrogates or density estimators can learn uncertainties on the network prediction, for instance, as Bayesian networks (BNNs) [68–70], repulsive ensembles (REs) [71–73], or likelihood methods [74, 75]. Even if the learned uncertainty is not used in a downstream analysis, a reliable uncertainty estimate is key to the justification for using an ML surrogate.

Our application is surrogate amplitudes encoded in modern neural networks [12–22], which ensures that higher-order scattering amplitudes can be evaluated as fast as tree-level amplitudes. The challenge for such surrogates is that perturbative quantum field theory requires them to reliably reproduce the theory prediction encoded in the training data. On top of the stringent accuracy requirements, we also study the reliability of the neural network surrogate via a learned uncertainty. In Sec. 2, we introduce

1. deterministic networks with a heteroscedastic loss to learn systematic uncertainties;

2. a Bayesian network to learn statistical and systematic uncertainties; and

3. repulsive ensembles to learn statistical uncertainties.

Here, we define statistical uncertainties as vanishing for infinite amounts of training data, while systematics can have many sources related to the training data, the network architecture, or the network training. Since the latter is the limiting factor in LHC applications, where networks are trained on simulations, we are typically interested in learned systematics, nevertheless testing that statistical limitations from limited training data are negligible.

In Sec. 3, we introduce our dataset of loop-induced amplitudes for the partonic process $gg \to \gamma\gamma g$ [14, 16], and in Sec. 4, we perform a systematic study of the effect of activation functions on the accuracy of training process by comparing static activation functions against learned ones. The latter employs two variants of Kolmogorov-Arnold networks (KANs) [76, 77]. KANs have recently been proposed as an alternative to normal feed-forward networks featuring improved accuracy and scalability, making them a suitable architecture for precision physics. In Sec. 5, we look at learned uncertainties induced by artificial noise on the amplitudes, by limited network expressivity or size, and through modern network architectures. For all cases and all three networks we can use pull distributions to show that the systematic uncertainties are calibrated. Finally, we have a brief look at the calibration of the learned statistical uncertainties in Sec. 6.

## 2 Learned uncertainties

In this study, we train a regression network to learn the transition amplitude or matrix element squared $A(x)$ as a function of phase space $x$,

$$A_{\text{NN}}(x) \approx A_{\text{true}}(x). \tag{1}$$

The key property of a transition amplitude is that $A_{\text{true}}$ is exact at a given order in perturbation theory, i.e. there are no stochastic processes entering the prediction. The training data $D_{\text{train}}$ consists of phase space points, the four-momenta of the scattering particles, and their corresponding amplitude values $(x, A)_j$. We continue with the derivation of the loss functions which allow us to provide learned uncertainties. We specialize the notation to the amplitude regression use case, however, the following subsections generalize to any regression problem.

### 2.1 Heteroscedastic loss

Denoting the network parameters as $\theta$, the network training maximizes the probability of the network parameters to describe the training data, $p(\theta|D_{\text{train}})$. Because we do not have access to this probability, we use Bayes theorem and instead minimize the negative log-likelihood on a set of amplitudes

$$\mathcal{L} = -\Big\langle \log p(A|x, \theta) \Big\rangle_{x \sim D_{\text{train}}}. \tag{2}$$

The form of the log-likelihood depends on the nature of the training data. For many applications, including those with systematic uncertainties, a natural choice is a Gaussian likelihood with heteroscedastic variance. It leads to a loss with two functions over phase space, $A_{\text{NN}}(x)$ and $\sigma(x)$

$$\mathcal{L}_{\text{het}} = \left\langle \frac{|A_{\text{true}}(x) - A_{\text{NN}}(x)|^2}{2\sigma(x)^2} + \log \sigma(x) \right\rangle_{x \sim D_{\text{train}}}. \tag{3}$$

Unlike for a fit, the variance $\sigma^2(x)$ is not known and has to be learned. This is possible if we include the normalization, so the minimization of the Gaussian exponent will drive $\sigma(x)$ towards large values, while its explicit appearance from the normalization will prefer a small variance. For this work, we have tested that it is sufficient to assume a Gaussian likelihood, but the heteroscedastic loss can be extended to a Gaussian mixture model for $\sigma(x)$ [69, 78].

Even if we never use the learned uncertainty, it can be useful as a cutoff in the likelihood term. This means it allows the network to ignore aspects of the data that cannot be learned and instead focus on improving other features without overtraining.

An aspect the heteroscedastic loss does not capture is the uncertainty from the actual training. A holistic treatment of network uncertainties requires more than a heteroscedastic loss

term. We first introduce a BNN for the quantification of per-amplitude uncertainty and then describe the same uncertainties using repulsive examples.

## 2.2 Bayesian neural network

Bayesian networks [68, 79–81] (BNNs) encode the amplitude $A(x)$ and its full uncertainty $\sigma(x)$ over phase space. For the LHC, they can be used, e.g. , for amplitude regression [16], jet calibration [70], classification [69], and generative networks [27, 52, 82, 83]. We start with the assumption that the amplitude for a given phase space point is given by a probability distribution $p(A|x)$ with mean

$$\langle A\rangle(x) = \int \mathrm{d}A\, A\, p(A|x). \tag{4}$$

The network encodes this probability in weight configurations, conditional on the training data. We train the network using variational inference [84, 85], by approximating the true posterior $p(\theta|D_{\text{train}})$ with a tractable $q(\theta)$, such that

$$p(A|x) = \int \mathrm{d}\theta\, p(A|x,\theta) p(\theta|D_{\text{train}}) \approx \int \mathrm{d}\theta\, p(A|x,\theta) q(\theta). \tag{5}$$

We implement this approximation by minimizing the KL-divergence

$$
\begin{aligned}
\mathrm{KL}[q(\theta), p(\theta|D_{\text{train}})] &= \int \mathrm{d}\theta\, q(\theta) \log \frac{q(\theta)}{p(\theta|D_{\text{train}})} \\
&= \int \mathrm{d}\theta\, q(\theta) \log \frac{q(\theta) p(D_{\text{train}})}{p(\theta) p(D_{\text{train}}|\theta)} \\
&= \mathrm{KL}[q(\theta), p(\theta)] - \int \mathrm{d}\theta\, q(\theta) \log p(D_{\text{train}}|\theta) + \log p(D_{\text{train}}) \int \mathrm{d}\theta\, q(\theta).
\end{aligned}
\tag{6}
$$

The first term arises from the prior and can be viewed as a weight regularization. The second term is the negative log-likelihood, where the sampling function $q(\theta)$ generalizes the standard dropout and will, for instance, avoid overtraining. The third term simplifies to $\log p(D_{\text{train}})$ and gives the marginal likelihood or evidence of $D_{\text{train}}$. It is a constant with respect to $\theta$, so the BNN loss is

$$\mathcal{L}_{\text{BNN}} = \mathrm{KL}[q(\theta), p(\theta)] - \left\langle \log p(D_{\text{train}}|\theta) \right\rangle_{\theta \sim q(\theta)}. \tag{7}$$

We are free to choose the prior $p(\theta)$, so we follow common practice and choose independent Gaussians with a given width for each weight. Choosing $q(\theta)$ as factorized Gaussians allows us to compute the KL-divergence in Eq.(7) analytically and to approximate the expectation in the second term with samples efficiently. Even though we make a simple ansatz, the non-linearities in the neural network allow to model complex posteriors [68].

**Uncertainties**

To extract the uncertainty for $A(x)$, we rewrite Eq.(4) such that we sample over $\theta$ and define an expectation value and the corresponding variance

$$
\begin{aligned}
\langle A\rangle(x) &= \int \mathrm{d}\theta\, q(\theta) \bar{A}(x,\theta), \qquad \text{with} \qquad \bar{A}(x,\theta) = \int \mathrm{d}A\, A\, p(A|x,\theta), \\
\sigma_{\text{tot}}^2(x) &= \int \mathrm{d}\theta\, q(\theta) \left[ \overline{A^2}(x,\theta) - \bar{A}(x,\theta)^2 + \left(\bar{A}(x,\theta) - \langle A\rangle(x)\right)^2 \right] \\
&\equiv \sigma_{\text{syst}}^2(x) + \sigma_{\text{stat}}^2(x),
\end{aligned}
\tag{8}
$$

where $\overline{A^2}(x,\theta)$ is defined in analogy to $\overline{A}(x,\theta)$. The total uncertainty factorizes into two terms. The first,

$$\sigma_{\text{syst}}^2(x) \equiv \int d\theta\, q(\theta)\, \sigma(x,\theta)^2 = \int d\theta\, q(\theta) \Big[\overline{A^2}(x,\theta) - \overline{A}(x,\theta)^2\Big],$$

corresponds to the learned error in the heteroscedastic loss in Eq.(3). Given exact $A_{\text{true}}(x)$, it vanishes in the limit of arbitrarily well-known data and perfect network training

$$p(A|x,\theta) = \delta(A - A_{\text{true}}(x)) \qquad \Leftrightarrow \qquad \overline{A^2}(x,\theta) = A_{\text{true}}(x)^2 = \overline{A}(x,\theta)^2. \tag{9}$$

We will see that it approaches a plateau for large training datasets, so we refer to it as a systematic uncertainty—accounting for a noisy data or labels [70], limited expressivity of the network (structure uncertainty) [16], non-optimal network architectures in the presence of symmetries, non-smart choices of hyperparameters or any other sources of systematic uncertainty.

The second error is the $\theta$-sampled variance

$$\sigma_{\text{stat}}^2(x) = \int d\theta\, q(\theta) \left[\overline{A}(x,\theta) - \langle A\rangle(x)\right]^2. \tag{10}$$

It vanishes in the limit of perfect training leading to uniquely defined network weights $\theta_0$,

$$q(\theta) = \delta(\theta - \theta_0) \qquad \Leftrightarrow \qquad \langle A\rangle(x) = \overline{A}(x,\theta_0). \tag{11}$$

It represents a statistical uncertainty in that it vanishes in the limit of infinite training data.

For small training datasets, these two uncertainties cannot be easily separated, but we can separate them clearly for sufficiently large training datasets [69], where $\sigma_{\text{stat}}$ approaches zero, while the systematic error $\sigma_{\text{syst}}$ reaches a finite plateau. Usually, in LHC physics, we can make sure to use enough training data, so

$$\sigma_{\text{tot}}(x) \approx \sigma_{\text{syst}}(x) \gg \sigma_{\text{stat}}(x). \tag{12}$$

Correspondingly, we focus on the extraction and validation of learned systematics. While we adopt a physics-oriented definition of uncertainties, the computer science community uses a different terminology. Systematic uncertainties are termed "aleatoric" or "data uncertainties". In general, these contributions cannot be reduced by collecting more observations, e.g. because of intrinsic stochasticity of the measurement process. Instead, "epistemic" uncertainties are reducible and are related to the lack of knowledge about the correct network parameters. This definition corresponds to our treatment of statistical uncertainties.

## 2.3 Repulsive ensembles

An alternative way to compute the uncertainty on a network output is ensembles, provided we ensure that the uncertainty really covers the probability distribution over the space of network functions [2, 86]. This means we need to cover the global and the local structure of the loss landscape.

**Ensembles**

Ensembles of networks trained with different initializations can provide us with a range of possible training outcomes by converging to different local minima in the loss landscape. The general formula for the weight update from the configuration $\theta^t$ to $\theta^{t+1}$

$$\theta^{t+1} = \theta^t + \alpha \nabla_{\theta^t} \log p(\theta^t | D_{\text{train}}), \tag{13}$$

with a learning rate parameter $\alpha$, does not lead to an interaction between the members. This means that if several networks converge to the same minimum, this will lead to a posterior collapse, so the ensemble will not provide us with a posterior distribution. On the other hand, if they approach different local minima, they should provide us with a reasonable distribution of networks, i.e. constitute a representative sample from the correct weight posterior distribution.

**Weight-space density**

Repulsive ensembles (REs) have been used in particle physics, for example, in cases where the training data does not allow for a stable training of BNNs [72,73]. They modify the usual update rule for minimizing the log-probability $p(\theta^t|D_{\text{train}})$ by gradient descent. To ensure a proper posterior estimation, they introduce a repulsive term into the updated rule of Eq.(13), to force the ensemble to spread out around the (local) loss minimum. A repulsive kernel $k(\theta, \theta_j)$ should increase with the proximity of the ensemble member $\theta$ to all other members,

$$\theta^{t+1} = \theta^t + \alpha \nabla_{\theta^t} \log p(\theta^t|D_{\text{train}}) - \alpha \frac{\nabla_{\theta^t} \sum_{j=1}^{n} k(\theta^t, \theta_j^t)}{\sum_{i=1}^{n} k(\theta^t, \theta_i^t)} \,. \tag{14}$$

Throughout this work we choose a Gaussian kernel.

Using properties of ordinary differential equations describing the network training as a time evolution, the extended update rule leads to an ensemble of networks sampling the posterior probability, $\theta \sim p(\theta|D_{\text{train}})$. To show this, we relate the extended update rule, or the discretized $t$-dependence of a weight vector $w(t)$, to a time-dependent probability density $\rho(\theta, t)$. Just as in the setup of conditional flow matching networks [2,27], we can describe this time evolution, equivalently, through an ODE or a continuity equation,

$$\frac{d\theta}{dt} = v(\theta, t), \qquad \text{or} \qquad \frac{\partial \rho(\theta, t)}{\partial t} = -\nabla_\theta \left[ v(\theta, t) \rho(\theta, t) \right]. \tag{15}$$

For a given velocity field $v(\theta, t)$ the individual paths $\theta(t)$ describe the evolving density $\rho(\theta, t)$ and the two conditions are equivalent. If we choose the velocity field as

$$v(\theta, t) = -\nabla_\theta \log \frac{\rho(\theta, t)}{\pi(\theta)}, \tag{16}$$

the two equivalent conditions read

$$\frac{d\theta}{dt} = -\nabla_\theta \log \frac{\rho(\theta, t)}{\pi(\theta)},$$
$$\frac{\partial \rho(\theta, t)}{\partial t} = -\nabla_\theta \left[ \rho(\theta, t) \nabla_\theta \log \pi(\theta) \right] + \nabla_\theta^2 \rho(\theta, t). \tag{17}$$

The continuity equation becomes the Fokker-Planck equation, for which $\rho(\theta, t) \to \pi(\theta)$ is the unique stationary probability distribution.

Next, we relate the ODE in Eq.(17) to the update rule for repulsive ensembles, Eq.(14). The discretized version of the ODE is

$$\frac{\theta^{t+1} - \theta^t}{\alpha} = -\nabla_{\theta^t} \log \frac{\rho(\theta^t)}{\pi(\theta^t)} \,. \tag{18}$$

If we do not know the density $\rho(\theta^t)$ explicitly, we can approximate it as a superposition of kernels,

$$\rho(\theta^t) \approx \frac{1}{n} \sum_{i=1}^{n} k(\theta^t, \theta_i^t), \qquad \text{with} \qquad \int d\theta^t \rho(\theta^t) = 1 \,. \tag{19}$$

We can insert this kernel approximation into the discretized ODE and find

$$\frac{\theta^{t+1} - \theta^t}{\alpha} = \nabla_{\theta^t} \log \pi(\theta^t) - \frac{\nabla_{\theta^t} \sum_i k(\theta^t, \theta_i^t)}{\sum_i k(\theta^t, \theta_i^t)} \, . \tag{20}$$

This form can be identified with the update rule in Eq.(14) by setting $\pi(\theta) \equiv p(\theta|D_{\text{train}})$, which means that the update rule will converge to the correct probability.

**Function-space density**

Applying a repulsion term in weight space has however a shortcoming for large neural networks. For instance, two networks with different weight configurations will be largely unaffected by a repulsive force in weight space although it is possible, especially for complex networks, that they encode the same function. We therefore introduce a repulsive term in the space of network outputs $A_\theta(x)$,

$$\frac{A_\theta^{t+1} - A_\theta^t}{\alpha} = \nabla_{A^t} \log p(A_\theta^t|D_{\text{train}}) - \frac{\sum_j \nabla_{A_\theta^t} k(A_\theta^t, A_{\theta,j}^t)}{\sum_j k(A_\theta^t, A_{\theta,j}^t)} \, . \tag{21}$$

The network training is still defined in weight space, so we have to translate the function-space update rule into weight space,

$$\frac{\theta^{t+1} - \theta^t}{\alpha} = \nabla_{\theta^t} \log p(\theta^t|D_{\text{train}}) - \frac{\partial A^t}{\partial \theta^t} \frac{\sum_j \nabla_A k(A_{\theta^t}, A_{\theta_j^t})}{\sum_j k(A_{\theta^t}, A_{\theta_j^t})} \, . \tag{22}$$

Because we cannot evaluate the repulsive kernel in the function space, we evaluate the function for a finite batch of points $x$,

$$\frac{\theta^{t+1} - \theta^t}{\alpha} \approx \nabla_{\theta^t} \log p(\theta^t|D_{\text{train}}) - \frac{\sum_j \nabla_{\theta^t} k(A_{\theta^t}(x), A_{\theta_j^t}(x))}{\sum_j k(A_{\theta^t}(x), A_{\theta_j^t}(x))} \, . \tag{23}$$

Finally, we turn the modified update rule into a loss function for the repulsive ensemble training. For a training dataset of size $N$, evaluated in batches of size $B$, $A_{\theta^t}(x)$ is understood as evaluating the function for $x_1, \ldots, x_B$. In the modified update rule, not all occurrences of $\theta$ are inside the gradient, so we use a stop-gradient operation, denoted as $\hat{A}_{\theta_j}(x)$. This gives us the loss function for $n$ repulsive ensembles

$$\mathcal{L}_{\text{RE}} = \sum_{i=1}^n \left[ -\frac{1}{B} \sum_{b=1}^B \log p(A|x_b, \theta_i) + \frac{\beta}{N} \frac{\sum_{j=1}^n k(A_{\theta_i}(x), \hat{A}_{\theta_j}(x))}{\sum_{j=1}^n k(\hat{A}_{\theta_i}(x), \hat{A}_{\theta_j}(x))} + \cdots \right] \, . \tag{24}$$

Strictly, the hyperparameter scaling the strength of the kernel should be chosen as $\beta = 1$. The dots indicate additional terms, for instance a weight normalization. A typical choice for the kernel is a Gaussian with a width given by the median heuristic [87]. As the output of the network contains the learned variance as well, we have to select in which space we apply the repulsion. We use the log-likelihood space. Applying the repulsion to the amplitudes and variances separately does not lead to different results.

## 2.4 Calibration and pulls

To determine if the learned uncertainty is correctly calibrated, we can look at pull distributions. Let us start with a deterministic network learning $A_{\text{NN}}(x) \approx A_{\text{true}}(x)$ using a heteroscedastic

loss. Given the learned local uncertainty $\sigma(x)$, we can evaluate the pull of the learned combination as

$$t_{\text{het}}(x) = \frac{A_{\text{NN}}(x) - A_{\text{true}}(x)}{\sigma(x)} \,. \tag{25}$$

If $\sigma(x)$ captures the absolute value of the deviation of the learned function from the truth exactly and for any $x$-value the pull would be

$$A_{\text{NN}}(x) = A_{\text{true}}(x) \pm \sigma(x) \qquad \Longleftrightarrow \qquad t(x) = \pm 1 \,. \tag{26}$$

For a stochastic source of uncertainties, the learned values $A_{\text{NN}}(x)$ will follow a Gaussian distribution around the mean $\langle A_{\text{NN}}(x) \rangle$. The width of this Gaussian should be given by the learned uncertainty, which means the pull will follow a unit Gaussian.

As an example, let us assume that we learn $A_{\text{true}}(x)$ from noisy training data,

$$A_{\text{true}}(x) \to A_{\text{train}}(x), \qquad \text{with} \qquad p(A_{\text{train}}(x)) = \mathcal{N}(A_{\text{true}}(x), \sigma^2_{\text{train}}(x)) \,. \tag{27}$$

If the network were allowed to overfit perfectly, the outcome would be $A_{\text{NN}}(x) = A_{\text{train}}(x)$. In that case, $\sigma(x)$ is not needed for the loss minimization and hence not learned at all. If we keep the network from overtraining, the best the network can learn from unbiased data is

$$A_{\text{NN}}(x) \approx A_{\text{true}}(x), \qquad \text{and} \qquad \sigma(x) \approx \sigma_{\text{train}}(x) \,. \tag{28}$$

In that case, we can define a pull function over phase space as

$$t_{\text{het}}(x) = \frac{A_{\text{NN}}(x) - A_{\text{train}}(x)}{\sigma(x)} \,. \tag{29}$$

It follows a unit Gaussian when the input noise is learned as the network uncertainty. Note that the pull compares the learned function with the training data, not with the idealized data in the limit of zero noise.

**Systematic pull**

We can translate this result for a deterministic network with a heteroscedastic loss to the BNN and variational inference in the limit $q(\theta) = \delta(\theta - \theta_0)$,

$$\langle A \rangle(x) = \int d\theta \, dA \, A \, p(A|x, \theta) \, q(\theta) = \int dA \, A \, p(A|x, \theta_0) \,. \tag{30}$$

The network is trained well if

$$p(A|x, \theta_0) \approx p(A|x) \,. \tag{31}$$

In the Gaussian case, this is equivalent to learning the mean and the width

$$\langle A \rangle(x) \approx A_{\text{true}}(x), \qquad \text{and} \qquad \sigma_{\text{syst}}(x) \approx \sigma_{\text{train}}(x) \,. \tag{32}$$

As argued above, the systematic pull relates the learned $A(x)$ to the actual and potentially noisy training data $A_{\text{train}}(x)$,

$$
\begin{aligned}
t_{\text{syst}}(x) &= \frac{1}{\sigma_{\text{syst}}(x)} \int dA \, [A - A_{\text{train}}(x)] \, p(A|x, \theta_0) \\
&= \frac{1}{\sigma_{\text{syst}}(x)} \left[ \int dA \, A \, p(A|x, \theta_0) - A_{\text{train}}(x) \int dA \, p(A|x, \theta_0) \right] \\
&= \frac{\langle A \rangle(x) - A_{\text{train}}(x)}{\sigma_{\text{syst}}(x)} \,.
\end{aligned}
\tag{33}
$$

Notably, we do not have to approximate the integral in the definition of $t_{\text{syst}}(x)$ since the network directly predicts the parameters of $p(A|x, \theta_0)$.

**Statistical pull**

To define a pull to test the calibration of the statistical uncertainty, we remind ourselves that we need to extract the statistical uncertainties defined in Eq.(10) using the unbiased sample variance from $N$ amplitudes sampled from the posterior weight distributions, $\theta_i \sim q(\theta)$,

$$
\begin{aligned}
\langle A \rangle(x) &\approx \frac{1}{N} \sum_i \overline{A}(x, \theta_i), \\
\sigma_{\text{stat}}^2(x) &\approx \frac{1}{N-1} \sum_i \left[ \overline{A}(x, \theta_i) - \langle A \rangle(x) \right]^2.
\end{aligned}
\tag{34}
$$

This definition provides us with the pull distribution

$$
\hat{t}_{\text{stat}}(x, \theta) = \frac{\overline{A}(x, \theta) - \langle A \rangle(x)}{\sigma_{\text{stat}}(x)}.
\tag{35}
$$

It samples amplitudes from the posterior, and it calculates the deviation from the expectation value $\langle A \rangle$ and the uncertainty $\sigma_{\text{stat}}(x)$, both computed by sampling the $\theta$. This pull is guaranteed to follow a standard Gaussian for large $N$, as $\sigma_{\text{stat}}(x)$ is calculated from the variance of $\overline{A}(x, \theta)$. Under the assumption of perfect training, we can replace the expectation value with its learning target, $\langle A \rangle(x) \approx A_{\text{true}}(x)$, and define the pull

$$
t_{\text{stat}}(x, \theta) = \frac{\overline{A}(x, \theta) - A_{\text{true}}(x)}{\sigma_{\text{stat}}(x)}.
\tag{36}
$$

This pull should again be a standard Gaussian, which means we can use it to test the calibration of the learned statistical uncertainty $\sigma_{\text{stat}}(x)$.

**Scaled pull**

The problem with Eq.(36) is that the pull depends on $\theta$, which means it can only be computed for individual members of the BNN or RE, but not for the sampled set of amplitudes. For a global pull, we would need to replace $\overline{A}(x, \theta)$ with a prediction after sampling and averaging over $\theta$.

Starting with systematics, when we evaluate the ensemble members together, the ensembling improves the central value, but the learned uncertainty does not benefit from the ensemble structure. In the next section, we will indeed see that ensembling, without or with repulsive kernel, will lead to a poorly calibrated, under-confident systematic pulls. Moving on to statistical uncertainties, when extracting the expectation value and the variance from $M$ samples, we know the scaling. Assuming identically distributed variables with unit weights, taking means for the prediction should reduce the statistical error by a factor $\sqrt{M}$, so we define the scaled standard deviation as

$$
\sigma_{\text{stat},M}(x) = \frac{\sigma_{\text{stat}}(x)}{\sqrt{M}}.
\tag{37}
$$

In analogy to Eq.(35), we can define the corresponding averaged pull as

$$
\hat{t}_{\text{stat},M}(x) = \frac{\langle A \rangle_M(x) - \langle A \rangle(x)}{\sigma_{\text{stat},M}(x)}.
\tag{38}
$$

To test the calibration, we would again replace the full expectation value with its learning target, $\langle A \rangle(x) \approx A_{\text{true}}(x)$ and define

$$
t_{\text{stat},M}(x) = \frac{\langle A \rangle_M(x) - A_{\text{true}}(x)}{\sigma_{\text{stat},M}(x)}.
\tag{39}
$$

Under the above conditions, it should follow a a standard Gaussian. By varying $M$, we can interpolate between the member-wise pull of Eq.(36) for $M = 1$ and the fully sampled pull with maximal $M$. We will see later that it starts deviating significantly beyond the single-element case $M = 1$. This does not imply that the statistical uncertainties are poorly calibrated but that the two samplings are not (sufficiently) independent.

# 3 Amplitude data and network architectures

As the benchmark for our surrogate amplitudes we choose the squared loop-induced matrix elements for the partonic process [14, 16]

$$gg \to \gamma\gamma g, \tag{40}$$

where we generate unweighted events with SHERPA [88], and then obtain the corresponding amplitudes with the NJET library [89]. A set of basic cuts on the partons,

$$p_{T,j} > 20 \text{ GeV}, \qquad |\eta_j| < 5, \qquad R_{jj,j\gamma,\gamma\gamma} > 0.4, \qquad p_{T,\gamma} > 40, 30 \text{ GeV}, \qquad |\eta_\gamma| < 2.37, \tag{41}$$

mimics the detector acceptance and object definition. The total size of the dataset is 1.1M phase space points with their corresponding amplitudes. Unless explicitly mentioned, we only use 70% of the data for training and train for 1000 epochs.

The accuracy of the network prediction is measured locally by

$$\Delta(x) = \frac{A_{\text{NN}}(x) - A_{\text{true}}(x)}{A_{\text{true}}(x)}. \tag{42}$$

To illustrate the accuracy of the networks, we histogram these values for a test dataset containing 20% of the complete dataset. The remaining 10% is used for validation and the selection of the best network. The width of the histogrammed $\Delta$-values for the test dataset gives the accuracy of the surrogate amplitude.

**Neural networks**

The following sections include detailed studies on the accuracy and learned uncertainties of several network architectures. With increasing complexity, we use the following network architectures:

- a fully connected network with linear layers followed by non-linearities (MLP);

- an MLP-I, like the MLP but including Mandelstam invariants as additional input features;

- a Deep Sets (DS) network [90], which learns an embedding for each particle type;

- a Deep Sets Invariants (DSI) network, i.e. DS with Mandelstam invariants as additional inputs;

- a Lorentz Geometric Algebra Transformer (L-GATr) network, a fully Lorentz and permutation-equivariant network architecture [18, 20].

In the following, the MLP networks are used for the BNN and RE results. The DS network introduces an embedding step before a standard MLP. It is shared across particles of the same type, and it is implemented as a fully connected network with a final representation vector of dimension 64. All the representations are concatenated before being passed to the second fully connected network, which predicts the amplitudes. The DSI version uses a structure similar to

the DS, where the input four-vectors are concatenated to Lorentz invariants. It is a specialized architecture for LHC amplitudes [18], combining a deep sets architecture [90] — providing permutation invariance — with Lorentz invariants [18].

As mentioned above, for both the MLP-I and the DSI, we augment all possible combinations of Mandelstam invariants from the input four-vectors,

$$s_{ij} = (p_i + p_j)^2 = 2p_i p_j \,, \tag{43}$$

which we additionally transform with a logarithm to obtain $\mathcal{O}(1)$ quantities that are easier to handle by neural networks. The L-GATr architecture can be used for amplitude regression [18, 20, 22, 73]. We use a modified version of the original network, with the addition of the heteroscedastic loss and the learned systematic uncertainties. The choice of hyperparameters for all the networks — if not stated explicitly in the text — can be found in App. A.

## 4  KAN amplitudes

Non-linearities are essential for building predictive models and thereby affect the accuracy of the learned amplitude surrogates. Here, we perform a systematic study of the effect of different types of non-linearities. In addition to comparing various fixed non-linear activation functions, we test learning non-linearities as part of the network training.

This can be realized using Kolmogorov-Arnold networks (KANs) [76, 77], which replace the fixed activation functions of an MLP by a non-linear learnable function for each graph edge. Moreover, it has been argued in Ref. [76], that KANs offer better accuracy and scalability.

KANs are based on the Kolmogorov-Arnold representation theorem, which also underlies the deep sets architecture [90] in a slightly different form. The Kolmogorov-Arnold theorem states that any multivariate smooth function with an $n$-dimensional input $x$ — $f : [0, 1]^n \to \mathbb{R}$ — can be written as a finite composition of uni-variate functions and addition,

$$f(x) = f(x_1, \ldots, x_n) = \sum_{q=1}^{2n+1} \Phi_q \left( \sum_{p=1}^{n} \phi_{q,p}(x_p) \right) , \tag{44}$$

where $\phi_{q,p} : [0, 1] \to \mathbb{R}$ and $\Phi_q : \mathbb{R} \to \mathbb{R}$. This implies that the only truly multivariate function is addition.

In practice, this decomposition is not very useful, since the appearing uni-variate functions often need to be non-smooth and/or fractal [91]. As shown in Refs. [76], this problem can be resolved by expanding the decomposition into multiple layers.

**Kolmogorov-Arnold networks**

A KAN network is built out of $L$ layers which we index using $l = 0, \ldots, L - 1$. The input of each layer is the $n_l$ dimensional vector $x_l$. Then, the action of the layer $l$ is defined via

$$x_{l+1,j} = \sum_{i=1}^{n_l} \phi_{l,j,i}(x_{l,i}), \qquad \text{with} \qquad j = 1, \ldots, n_{l+1}, \tag{45}$$

where the $n_{l+1} \times n_l$ $\phi_{l,j,i}$ functions are non-linear and learnable. These could be splines or rational functions [76, 92, 93]. Alternatively, the action of the layer $l$ can be written using the

operator matrix $\Phi_l$,

$$
x_{l+1} = \underbrace{\begin{pmatrix} \phi_{l,1,1}(\cdot) & \phi_{l,1,2}(\cdot) & \cdots & \phi_{l,1,n_l}(\cdot) \\ \phi_{l,2,1}(\cdot) & \phi_{l,2,2}(\cdot) & \cdots & \phi_{l,2,n_l}(\cdot) \\ \vdots & \vdots & \ddots & \vdots \\ \phi_{l,n_{l+1},1}(\cdot) & \phi_{l,n_{l+1},2}(\cdot) & \cdots & \phi_{l,n_{l+1},n_l}(\cdot) \end{pmatrix}}_{\equiv \Phi_l} x_l \, ,
\tag{46}
$$

where the function $\phi_{l,j,i}$ takes $x_{l,i}$ as input. Using this layer definition, the whole KAN network can be written in the form

$$
\text{KAN}(x) = (\Phi_{L-1} \circ \Phi_{L-2} \circ \ldots \circ \Phi_1 \circ \Phi_0) x \, .
\tag{47}
$$

For comparison, a normal MLP network has the form

$$
\text{MLP}(x) = (W_{L-1} \circ \text{activation} \circ W_{L-2} \circ \ldots \circ \text{activation} \circ W_0) x \, ,
\tag{48}
$$

where the $W_l$ are linear operations and "activation" stands for the chosen activation function. KANs have been previously applied to collider physics problems in Refs. [94, 95].

**Kolmogorov-Arnold layers using GroupKAN**

As an alternative in-between full KAN networks — with non-linear learnable functions for each graph edge — and normal MLP networks — with fixed activation functions —, GroupKAN networks have been introduced in Ref. [93]. They can be viewed as a normal MLP network with one or more learnable activation functions. This corresponds to replacing each activation layer with a GroupKAN layer,

$$
\text{activation}(x) \to \text{GroupKANlayer}_l(x) = \begin{pmatrix} \phi_{l,g_l(1)}(x_1) \\ \phi_{l,g_l(2)}(x_2) \\ \vdots \\ \phi_{l,g_l(n_l)}(x_{n_l}) \end{pmatrix} \, .
\tag{49}
$$

Here, the various functions are put into groups defined by the sub-indices

$$
g_l : \{1, 2, \ldots, n_l\} \to \{1, 2, \ldots, m_l\} \, ,
\tag{50}
$$

where $g_l(i) = k$ if $i$ belongs to group $k$ and $m_l$ is the number of groups in the layer $l$. The number of groups can vary between one — only one common function — to $n$ — all entries have a different activation function.

Then, the overall GroupKAN network can be written in the form

$$
\text{GroupKAN}(x) = (W_{L-1} \circ \text{GroupKANlayer}_{L-2} \circ W_{L-2} \circ \ldots \circ \text{GroupKANlayer}_0 \circ W_0) x \, .
\tag{51}
$$

Compared to the full KAN network, the GroupKAN reduces the complexity significantly. Per layer, only $m_l \le n_l$ functions need to be learned instead of $n_{l+1} \times n_l$ functions for the full KAN network. The GroupKAN is, however, still more expressive than a normal MLP. The GroupKAN layer can also be straightforwardly inserted in any other network architecture.

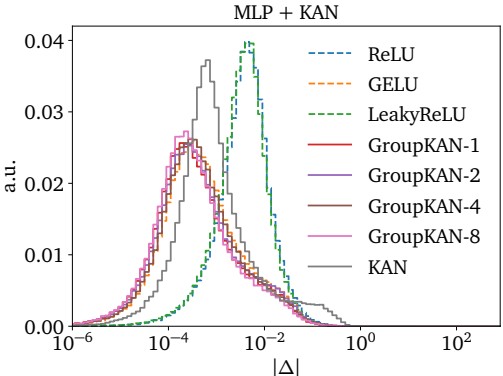
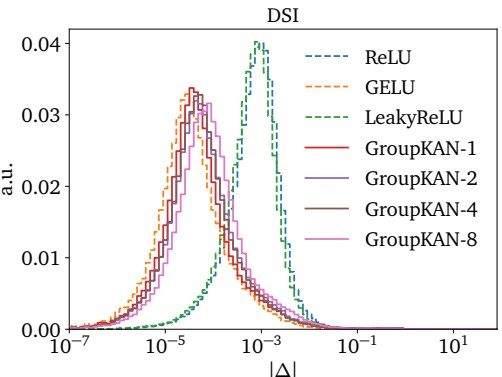

Figure 1: Accuracy on a logarithmic scale for the MLP, DSI, and KAN networks.

**Activation functions**

We investigate three different architectures. The first is a simple MLP with three hidden layers and 128 hidden dimensions. We compare the use of different activation functions. In particular, we compare several fixed functions from the rectified linear unit family — ReLU, GELU, leakyReLU — to a GroupKAN approach with 1, 2, 4, or 8 groups, which we denote as "GroupKAN-1" etc.. We parameterize the learnable GroupKAN activation functions using rational functions [92, 93] with an order five polynomial in the numerator and denominator, respectively. The MLP networks have $\sim 4 \cdot 10^4$ parameters. In addition to the MLP architecture, we test the DSI architecture. Also here, we compare different fixed activation functions to the GroupKAN approach. The DSI networks have $\sim 2 \cdot 10^5$ parameters. Moreover, we test a full KAN network with three hidden layers and 64 hidden dimensions. We use B-splines with a polynomial order of three for the learnable functions and a grid size of ten. The KAN network has $\sim 1 \cdot 10^5$ parameters.

We show $|\Delta|$ distributions for the various MLP and KAN networks in the upper left panel of Fig. 1. We see that the GroupKAN networks provide the most accurate predictions, with the bulk of the distribution lying between $10^{-4}$ and $10^{-3}$. We observe no significant performance increase when increasing the number of groups. The GELU network is slightly worse than the GroupKAN networks. In contrast, the ReLU and LeakyReLU networks have significantly worse performance with $|\Delta|$ values centered around $\sim 10^{-2}$. The KAN network lies in between the GroupKAN and ReLU/LeakyReLU results. Notably, the KAN distribution has a shoulder for large $|\Delta|$ indicating that the prediction is significantly worse for a subset of all amplitudes.

Next, we turn to the DSI architecture as shown in the right panels of Fig. 1. We again observe that the GroupKAN and the GELU networks have similar performance, while the ReLU and LeakyReLU networks perform significantly worse. In contrast to the MLP networks, the GELU network, however, slightly outperforms the GroupKAN networks.

We provide a visualization of the learned activation functions for the GroupKAN-1 networks in Fig. 2. In the left panel, the MLP GroupKAN-1 activation functions for the three layers are shown in comparison to the tested fixed activation functions. The learned activation functions behave similarly to the fixed activation functions for $x \sim 0$ but deviate significantly for larger $|x|$ values. The behavior is similar for the DSI GroupKAN-1 network, for which we show the activation layer of the summary net in the right panel of Fig. 2.

In conclusion, we find full KAN networks to provide no significant improvement over architectures with fixed activation functions. GroupKAN layers can, however, be a useful tool to enhance the accuracy of MLP networks. For more complex architectures like the DSI architecture, we find no improvement above a well-chosen fixed activation function. They, nevertheless, can be helpful in establishing a basis line if an extensive scan of different fixed activation functions is not feasible.

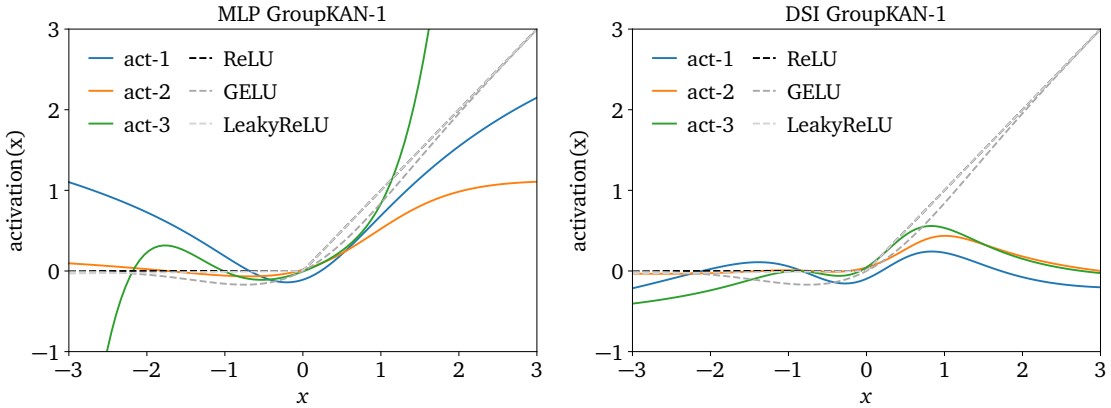

Figure 2: Learned activation functions for the GroupKAN-1 MLP and DSI networks.

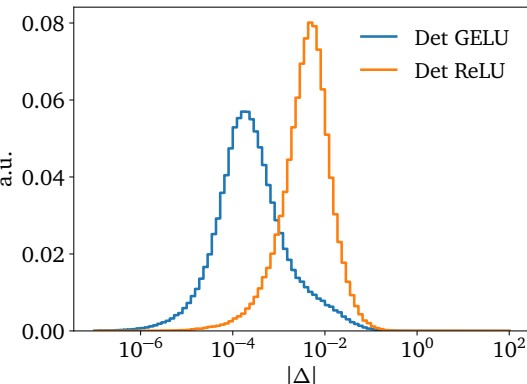

Figure 3: Comparing GELU and ReLU as possible activation functions for a deterministic network (Det).

From the MLP and DSI GroupKAN study, Fig. 2 shows that there is a slight preference towards choosing the GELU as an activation function compared to the ReLU. Therefore, we compared the impact of both activation functions on a deterministic network (Det). Figure 3 shows this comparison, indicating the benefit of using the GELU activation function compared to the ReLU one.

# 5 Systematics

From the two sources of uncertainties, statistics and systematics, we first study the systematics. They can be extracted through the heteroscedastic loss, which in turn can be included in the BNN and the REs. For LHC applications, like amplitude regression, it dominates the total uncertainty. In this section, we study the surrogate limitations traced by systematic limitations, starting with artificial noise and then moving on to the network expressivity and symmetry-aware network architectures.

Unless mentioned explicitly, we use a BNN prior variance $\sigma_{\text{prior}} = 1$ and a repulsive prefactor of $\beta = 1$ for the repulsive ensemble.

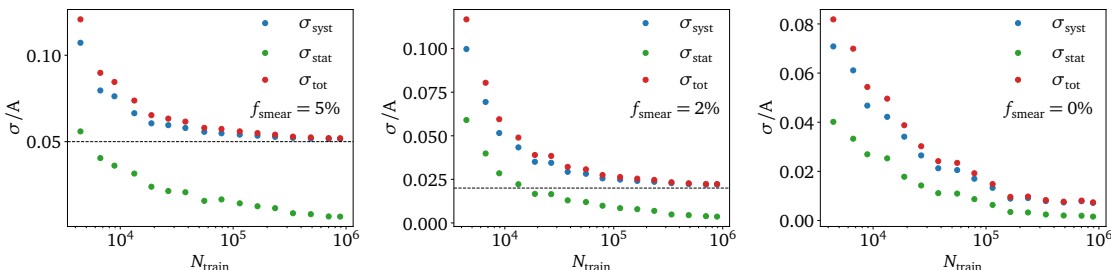

Figure 4: Relative systematic and statistical uncertainties learned by the BNN as a function of the dataset size, for 5%, 2%, and zero artificial noise on the training data.

## 5.1 Systematics from noise

There are many sources of systematic uncertainties, and as a starting point, we apply Gaussian noise to the data set to determine if the noise can be learned by the networks [70]. Using Eq.(27), we define the artificial noise level relative to the amplitude,

$$A_{\text{train}} \sim \mathcal{N}(A_{\text{true}}, \sigma_{\text{train}}^2), \qquad \text{with} \qquad \sigma_{\text{train}} = f_{\text{smear}} A_{\text{true}}, \tag{52}$$

where we consider the relative noise fractions

$$f_{\text{smear}} = \{0.25, 0.5, 0.75, 1, 2, 3, 5, 7, 10\}\%. \tag{53}$$

This introduces stochastic systematics in the data. Assuming that the additional systematics factorizes, it should appear as added to the other uncertainties in quadrature,

$$\sigma_{\text{tot}}^2 = \sigma_{\text{syst}}^2 + \sigma_{\text{stat}}^2 = \sigma_{\text{syst},0}^2 + \sigma_{\text{noise}}^2 + \sigma_{\text{stat}}^2 \quad \Leftrightarrow \quad \sigma_{\text{noise}}^2 = \sigma_{\text{syst}}^2 - \sigma_{\text{syst},0}^2. \tag{54}$$

The contribution $\sigma_{\text{syst},0}$ represents the systematic uncertainty stemming, for instance, from a limited network expressivity.

In Fig. 4 we show the two learned uncertainties from the BNN as a function of the amount of training data. Each point is the median of the respective uncertainty extracted from a test sample of 200K amplitudes. We perform this scan for the noise levels $\sigma_{\text{smear}} = \{5, 2, 0\}\%$ (left to right). For the largest noise rate, we see that the learned statistical uncertainty vanishes towards the large training sample, leaving us with a well-defined plateau for the systematic uncertainty. On the other hand, we also see that the split into statistical and stable systematic uncertainties only applies in the limit of large training samples and vanishing $\sigma_{\text{stat}}$. For finite training data size, the systematic uncertainty depends on the training sample size as well. The reason is that for limited training data, systematic uncertainties can occur when the, in principle, sufficient expressivity of the network is not used after limited training.

When we reduce the artificial noise to 2%, the learned systematics plateau drops to the corresponding values, and without added noise, it reaches a finite plateau below 1% relative systematics. It represents the next level of systematic uncertainty. We will see in Sec. 5.2 that it is related to the expressivity or size of the regression network.

In Fig. 5 we confirm that the learned systematics indeed reproduce the artificial noise added to the training data. In the left panel, we show the extracted uncertainties for the BNN, the REs, and a deterministic network with a heteroscedastic loss. All three methods agree for the systematics, including the feature that for noise below 2% the learned systematics approach a new target value around 0.5% systematics.

For the BNN and the RE we also look at the learned statistical uncertainty, which should be independent of the added noise. For the BNN, the learned statistical uncertainty is below 0.1% in the no-noise limit. Adding noise makes this estimate less reliable, which reflects the

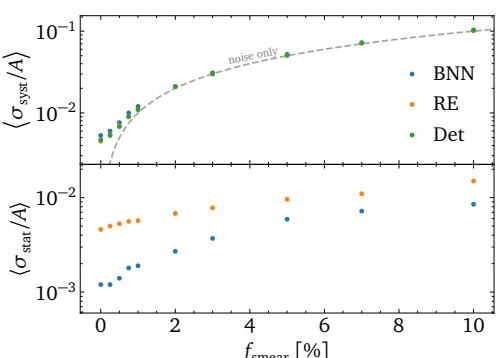
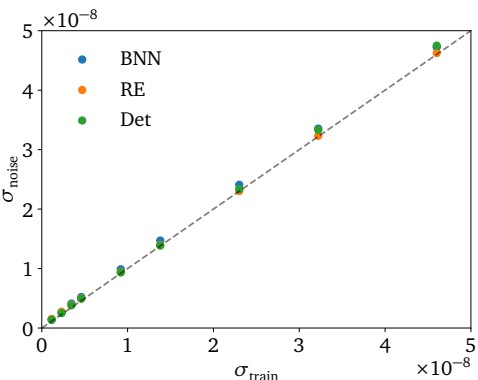

Figure 5: Left: relative uncertainty versus artificial noise for different network architectures. The "noise only" curve shows the scaling of the systematics, assuming the added noise is the only source of uncertainty. The exact numbers are given in Tab. 2. Right: extracted noise, defined in Eq.(54), as a function of the input noise for BNN, REs, and a heteroscedastic deterministic network.

numerical problem of separating two contributions and adding in quadrature if one of them is a factor 100 larger than the other. Interestingly, the statistical uncertainty learned by the REs is significantly larger. Because the data efficiency of the two network training can be different, this is expected, and we will discuss the learned statistical uncertainties more in Sec. 6

In the left panel of Fig. 5 we visualize the learned statistical and systematic uncertainties as a function of $f_{smear}$. As the noise tends to zero, only a systematic component from the limited network expressivity $\sigma^2_{syst,0}$ remains, observed as the deviation from the "noise only" dotted line. This is further discussed in Sec. 5.2. Instead, in the right panel of Fig. 5 we show the calibration curve for the input vs learned noise using the definition of Eq.(54). It confirms the excellent and consistent behavior of the three different implementations. The differences in the learned statistical uncertainties are numerically too small to affect the calibration significantly. As a word of caution — in the Appendix, we also show the extracted noise when we apply REs without a heteroscedastic loss. While one might speculate that in this setup, the REs would still learn the systematic uncertainty, it really does not. REs without heteroscedastic loss are really limited to statistical uncertainties.

The correlation shown in Fig. 5 indicates that the learned systematics extract the added noise correctly. However, the correlation only shows the median uncertainty, so we still want to check the uncertainty distributions using the systematic pulls introduced in Eq.(33). In Fig. 6 we show the relative accuracy and the systematic pull distributions for three different noise levels. If stochastic systematics are learned correctly, the pull should follow a unit Gaussian. In the upper panels, we first confirm that the accuracy improves with less noise, first consistently for the three methods, and towards zero noise with different accuracies for the REs on the one side and the BNN and the heteroscedastic noise. In all cases, the relative accuracy distributions follow approximate Gaussians.

In the second row Fig. 6 we show the systematic pull, combining the accuracy in the numerator with the learned uncertainty in the denominator. This combination should become a universal unit Gaussian. The BNN and the heteroscedastic network indeed reproduce this pattern for all added noise levels. In the limit of zero noise, the RE gives a too-narrow systematic pull distribution, indicating that the learned systematics from the heteroscedastic modification of the REs is too conservative. From Fig. 5 we can speculate that the too-large learned systematics in the limit of zero noise is related to the fact that the REs extract a common $\sigma_{syst} \approx \sigma_{stat} \approx 0.5\%$ in this limit. The heteroscedastic loss in the REs only extracts the correct systematics when it is larger than the learned statistical uncertainty.

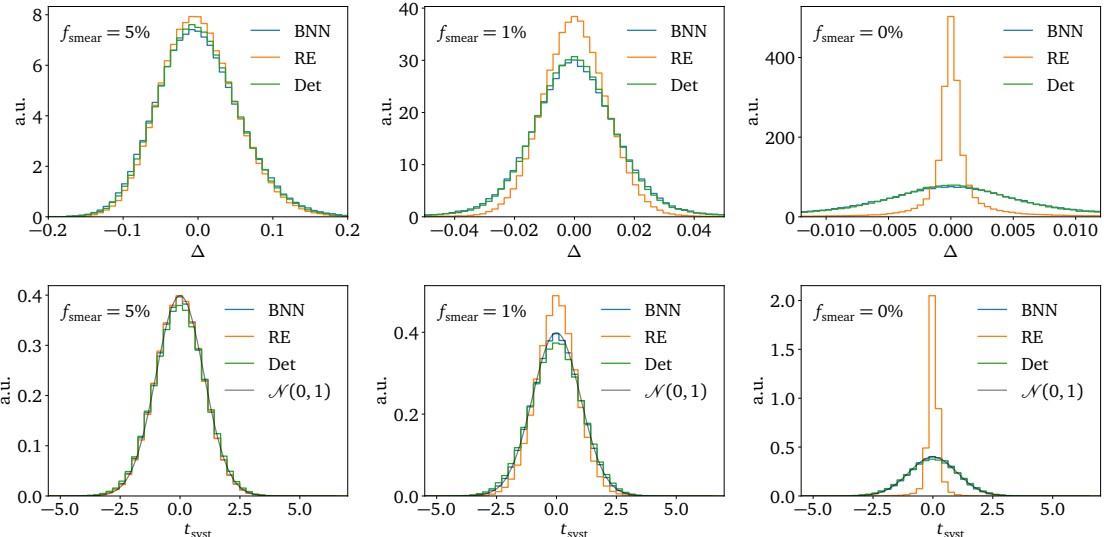

Figure 6: Relative accuracy (upper) and systematic pull (lower) for the BNN, REs, and heteroscedastic loss, for decreasing added noise.

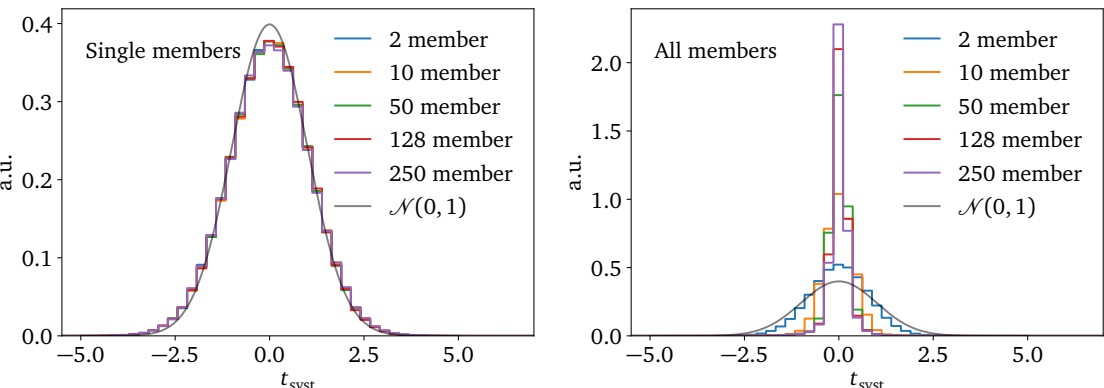

Figure 7: Systematic pulls for the repulsive ensemble with a different number of ensemble members. We show the systematic pulls from single ensemble members (left) and for the standard averaged prediction of the full ensemble (right).

## Calibrating ensembles

In Fig. 6 we observe a poor calibration of (repulsive) ensembles. We have checked that for our setup the repulsive kernel has hardly any impact, so we can look at standard ensembles to understand this issue.

In the left panel of Fig. 7 we confirm that the different initializations of the ensemble members lead to convergence in different local minima, which may predict some amplitudes better than others. Extracted independently for each member and with a heteroscedastic loss, we see that the pull is perfectly calibrated.

In Sec. 2.4 we have already discussed that ensemble training improves the accuracy of the amplitude regression, but the learned uncertainty does not benefit from them. This is confirmed by the right panel of Fig. 7, where the RE pull from an larger ensemble becomes increasingly too narrow. While the mean network prediction becomes more accurate, the learned $\sigma_{\text{syst}}$ only accounts for the systematics of the single network, which leads to the poorly calibrated systematic RE pull.

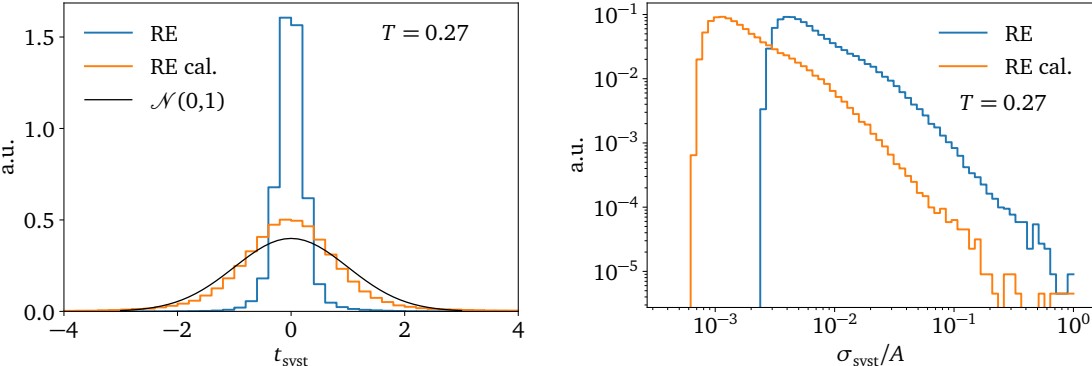

Figure 8: Systematic pulls (left) and $\sigma_{\text{syst}}/A$ (right) for the repulsive ensemble before and after the calibration. The fitted value of the temperature parameter is $T = 0.27$.

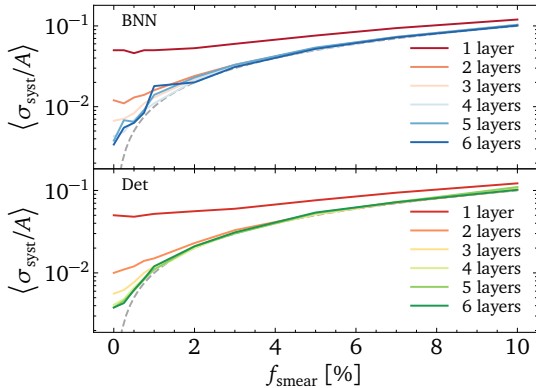

Figure 9: Relative uncertainty versus added noise for different numbers of hidden layers. We show results for a deterministic network with heteroscedastic loss and for the BNN. The exact numbers are given in Tab. 3.

Because the narrow pull distributions as Gaussian do not show any bias, we can solve this issue using a targeted calibration step. The simplest solution introduces a single global parameter, which rescales all the amplitudes and effectively changes the parameters of each learned Gaussian as $\sigma_{\text{syst}} \rightarrow \sigma_{\text{syst}} \times T$. Fig. 8 shows the systematic pull before and after the calibration procedure. The calibration parameter $T$ is estimated using stochastic gradient descent on the full training dataset and evaluated on the test dataset. The loss used for the optimizer is the usual heteroscedastic loss,

$$\mathcal{L}_T(x) = \left\langle \frac{|A_{\text{true}}(x) - \overline{A}(x)|^2}{2\sigma^2(x)T^2} + \log \sigma(x)T \right\rangle_{x \sim D_{\text{train}}}. \tag{55}$$

## 5.2 Systematics from network expressivity

Adding artificial noise as the, by definition, dominant uncertainty to our amplitude data immediately bears the question: where does the systematic uncertainty in the limit of no noise in Fig. 6 come from?

In this section, we look at the effect of the network size and network expressivity on the systematics. In Fig. 9 we show the learned systematics as a function of the noise and for different numbers of hidden layers. Starting with the deterministic network and a heteroscedastic

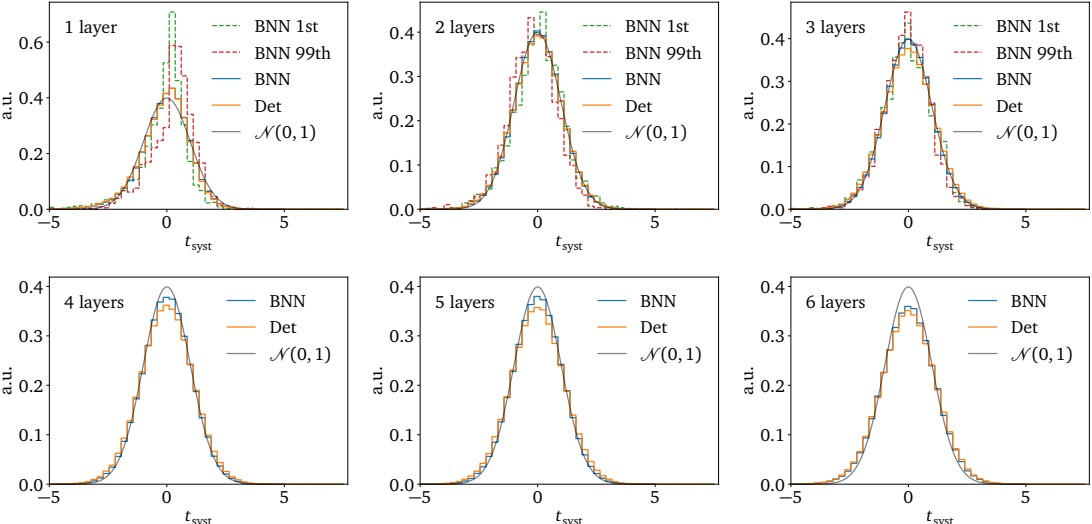

Figure 10: Systematic pulls for all test amplitudes without noise, shown for an increasing number of hidden layers. For one and two hidden layers we also show the extreme quantiles of the amplitudes are shown, to identify the failure mode.

loss, and without added noise, the median relative systematics decreases from 5% for one hidden layer to better than 0.5% for five or six hidden layers. On the other hand, we see that training more than three or four hidden layers is starting to be less stable. For just one hidden layer, all noise scenarios are not learned correctly. This indicates that the network is too small to extract the amplitudes and the uncertainty. This improves for two hidden layers, at least down to 2% noise, and for three hidden layers to 0.25% noise. This means that more expressive networks can describe the amplitudes with smaller and smaller noise, finally reaching a systematics plateau around $\langle \sigma_{\text{syst}}/A \rangle = 0.38\%$.

Also in Fig. 9 we repeat the same study for the BNN, which has to separate these joint systematics from the statistical uncertainty, according to Fig. 5 and Tab. 2 at the 0.1% level in the limit of little or no noise. We know that very large BNNs run into stability issues when the Bayesian layers destabilize the training. The reason for this is a too large deterministic network can switch off unused weights by setting them to zero. A BNN can only do this for the mean, while the widths of the network parameter will be driven to the prior hyperparameter. During training, these parameters with zero mean but finite width add unwanted noise. The solution is to only learn a per-weight variance for the number of layers needed to express the learned uncertainty, while assuming a delta distribution for the remaining weights. Specifically, we only use a Bayesian last layer for our networks with more than three hidden layers. With this caveat, the BNN results in Fig. 9 reproduce the results from the heteroscedastic loss, even with more stable training thanks to the BNN regularization.

In Fig. 10 we again show the systematic pull distributions, now as a function of the number of hidden layers. All pull distributions are close to the expected unit Gaussian for stochastic sources of the underlying systematics. Given that the systematics we are looking at is the increasing expressive power of the network, this Gaussian distribution is not guaranteed. For one hidden layer we also show the lowest and highest quantiles in the amplitude size separately. Both of them are driving the deviation of the pull distribution from the unit Gaussian, indicating that learning the extreme amplitude values challenges the network expressivity. From two hidden layers onward the situation is better. For large networks, the pull distributions of the BNN are becoming slightly too wide, which means the systematic uncertainty is underestimated, which can be explained by the small number of actual Bayesian network weights and the choice of the prior hyperparameter.

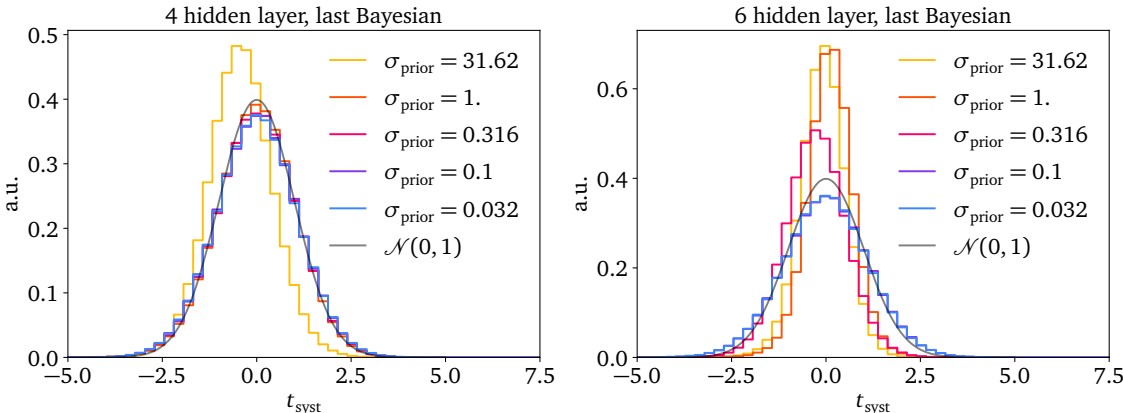

Figure 11: Systematic pulls for all test amplitudes without noise, shown for 4 and 6 hidden layers. We use a BNN with different prior widths, and only the last layer is Bayesian.

Table 1: Comparison of different architectures, starting from a standard heteroscedastic network and adding different features, then turning to deep sets without and with invariants.

| architecture | # hidden layers | | | | | |
|---|---|---|---|---|---|---|
| | 1 | 2 | 3 | 4 | 5 | 6 |
| $\langle \sigma_{\text{Det}}/A \rangle$ (Tab. 3) | 0.050 | 0.010 | 0.0056 | 0.0041 | 0.0038 | 0.0038 |
| $\langle \sigma_{\text{Det}}^{\text{I}}/A \rangle$ | 0.00380 | 0.00138 | 0.00098 | 0.00086 | 0.00102 | 0.00104 |
| $\langle \sigma_{\text{Det}}^{\text{I}}/A \rangle$ float64 | - | - | 0.00106 | - | - | 0.00107 |
| $\langle \sigma_{\text{Det}}^{\text{I}}/A \rangle$ float64+leakyReLU | - | - | 0.00091 | - | - | 0.00092 |
| $\langle \sigma_{\text{Det}}^{\text{DS}}/A \rangle$ | - | - | 0.00019 | - | - | - |
| $\langle \sigma_{\text{Det}}^{\text{DSI}}/A \rangle$ | - | - | 0.000054 | - | - | - |
| $\langle \sigma_{\text{Det}}^{\text{DSI}}/A \rangle$ with L2-norm | - | - | 0.000068 | - | - | - |
| $\langle \sigma_{\text{Det}}^{\text{DSI}}/A \rangle$ 2000 epochs | - | - | 0.000039 | - | - | - |
| $\langle \sigma_{\text{BNN}}^{\text{DSI}}/A \rangle$ | - | - | 0.000070 | - | - | - |
| $\langle \sigma_{\text{RE}}^{\text{DSI}}/A \rangle$ | - | - | 0.000051 | - | - | - |

In Fig. 11 we check the stability of the BNN with four and six hidden layers as a function of the prior hyperparameter. While limiting the BNN sampling to the last layer stabilizes the network, the question is if the trained network still samples the entire posterior. To see this, we study the systematic pulls for different values of the prior hyperparameter. We confirm the existence of a broad plateau for this hyperparameter but shift to smaller prior values for more hidden layers. Larger networks with a smaller fraction of Bayesian layers need to be pushed to sample the full statistical uncertainty.

## 5.3 Systematics from symmetries

After identifying added noise and the number of network parameters as the two leading sources of systematic uncertainties, the question is what source of systematics leads to the plateau value for the heteroscedastic network around 0.38% in Fig. 9 and Tab. 3.

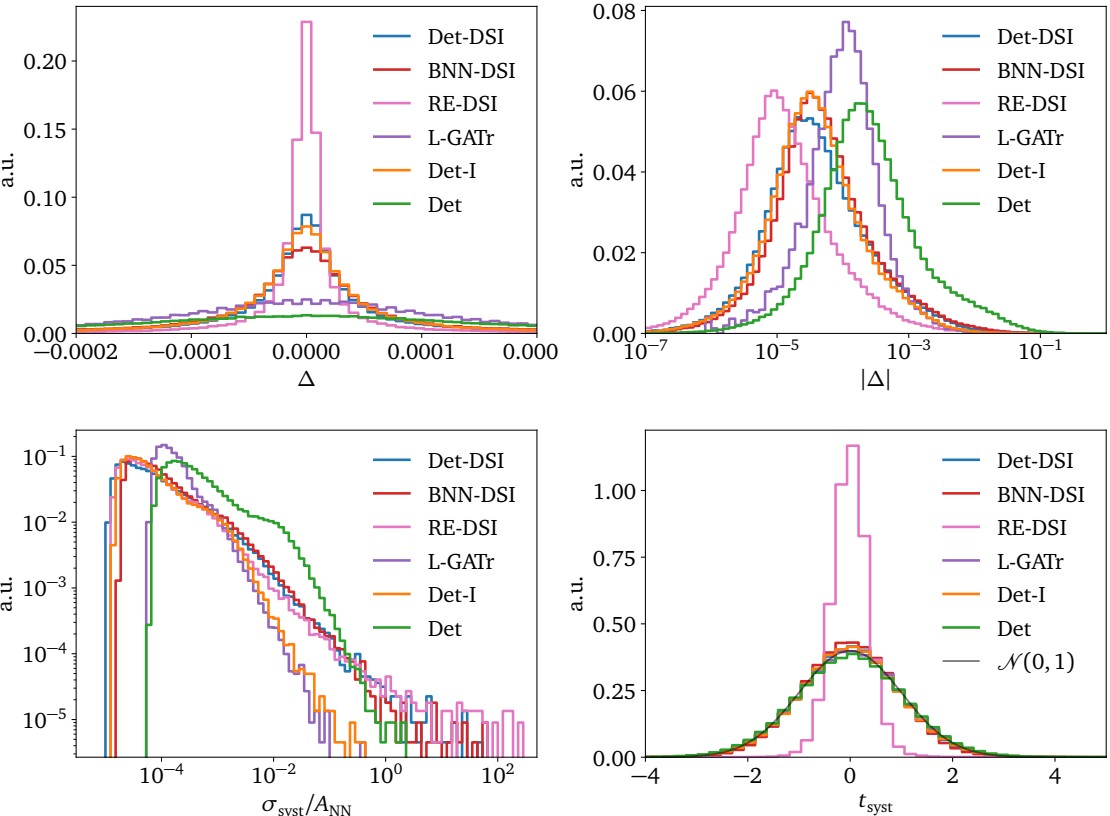

Figure 12: Upper: accuracy on a linear and logarithmic scale for different network architectures using a heteroscedastic loss, as well as the BNN and the RE version of the DSI network. Lower: relative systematic uncertainties and systematic pulls, indicating the poor calibration of $\sigma_{\text{syst}}$ learned by REs.

To identify this source, we work with more advanced architectures that incorporate the symmetries of our amplitude data. The two key symmetries for LHC amplitudes are Lorentz and permutation invariance [18,20], where for our simple $(2 \to 3)$ process Lorentz invariance is more relevant. In Tab. 1, we document the improvements when improving the network architecture. We start by considering the MLPI, which includes Mandelstam invariants as additional features, $\sigma^{\text{I}}$. This leads to a sizeable improvement in the learned uncertainty, which we know tracks the corresponding improvement in accuracy. This improvement exists for all network sizes, with a performance plateau from three to six hidden layers, and for three hidden layers it reduces the systematics to around 0.1%. Further changes, like higher machine precision or alternative activation functions, do not improve the performance of the standard MLP architecture.

In a second step shown in Tab. 1, we replace the standard MLP network with a deep sets architecture designed for amplitude regression, $\sigma^{\text{DS}}$. We find a significant performance boost to a relative systematic uncertainty around 0.02%. Combining the deep sets architecture with Lorentz invariants defines $\sigma^{\text{DSI}}$, resulting in another drop in the relative systematic uncertainty to around 0.005%. This level can be stabilized by adding an L2-normalization and training the normalized network for 2000 epochs. For the BNN version of the DSI network, the systematic uncertainty does not quite reach this level but comes extremely close. Notably, when switching to the DSI architecture, we have to train and evaluate the network in double precision to avoid numerical artifacts.

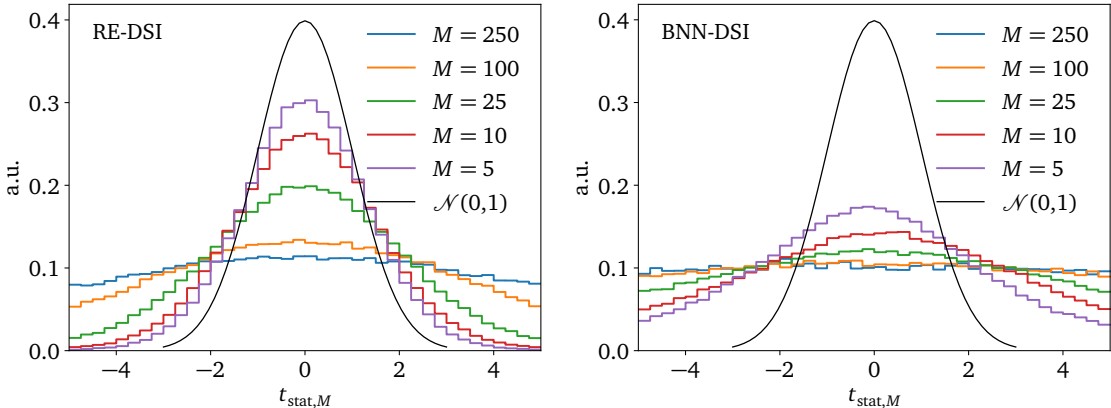

Figure 13: Distribution of $t_{\text{stat},M}(x)$, defined in Eq.(39), for a test set of amplitudes learned by a RE-DSI (left) and a BNN-DSI (right). The mean $\langle A \rangle$ and the statistical uncertainty $\sigma_{\text{stat}}$ are estimated from 512 amplitudes.

Finally, we again check the systematic pull of the advanced network architectures in Fig. 12. In the upper panels, we show the improved accuracy of the networks, now also adding the new Lorentz-equivariant geometric algebra transformer (L-GATr) [18, 20]. This architecture improves the scaling with the number of particles in the final state and is almost on par with the leading DSI, BNN-DSI, and uncalibrated RE-DSI variants. Their accuracy is stable on the $10^{-5}$ level, with suppressed and symmetric tails. The relative systematic uncertainty is asymmetric, but this is exactly the distribution we expect from a unit-Gaussian pull distribution, i.e. it reflects the asymmetric distributions of the training and test amplitudes. All networks, except for the RE variant, have perfectly calibrated systematic uncertainties. This calibration traces the next leading systematics at the $10^{-5}$ level, whatever it is.

## 6   Statistics

The second contribution to the uncertainty arises from limited statistics and it should vanish in the limit of infinite training data. In this section, we test the statistical pulls defined in Sec. 2.4 for both BNN and RE. We start by applying the scaled statistical pull to a large set of amplitudes from the approximate posterior and then turn to the sampled statistical pull. After validating the statistical pull, we confirm that the statistical uncertainties are a sub-leading contribution to the total error for our most precise network.

**Scaled statistical uncertainties**

We start from a $\theta$-independent pull, as defined in Eq.(39) and study its scaling behavior. For these results, we use $N = 512$ samples, which is the largest number of members we can train in parallel for the ensemble. The left panel of Fig. 13 shows the pulls for the RE-DSI. The scaling behavior we expect for $t_{\text{stat},M}$ agrees with a standard Gaussian pull for $M \ll N$. As $M$ approaches the number of samples used to estimate $\sigma_{\text{stat}}$, the correlation between the two variables increases until the scaling of $\sigma_{\text{stat},M}$ does not hold anymore, making the pull narrower. Removing the $\theta$ dependence in the $A_{\text{true}}$ residuals also does not provide good calibration, as with increasing $M$ the pulls drift towards over-confident uncertainties. We observe a similar behavior for the BNN-DSI in the right panel of Fig. 13. We conclude that the sample variance of the mean is not a reliable uncertainty and, for the rest of the section, shift to the sampled statistical pull with 128 members.

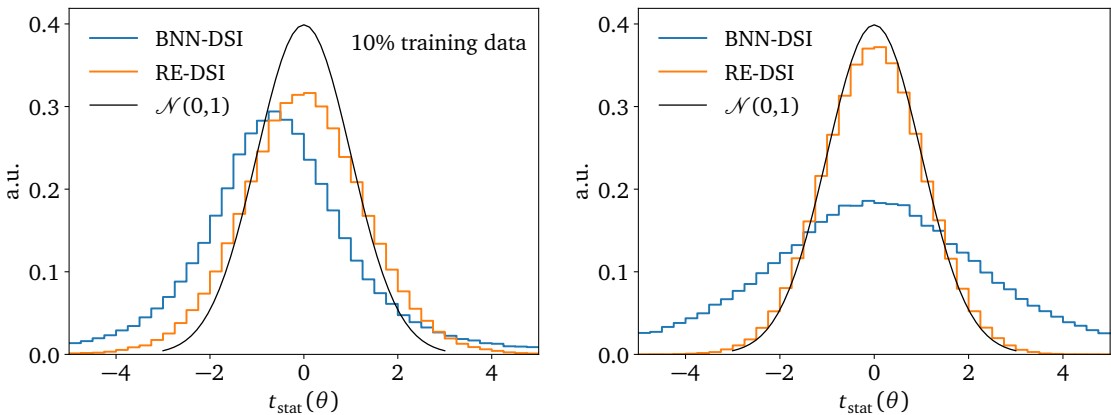

Figure 14: Statistical pulls from the exact training data for 10% (left) and all (right) of the training data.

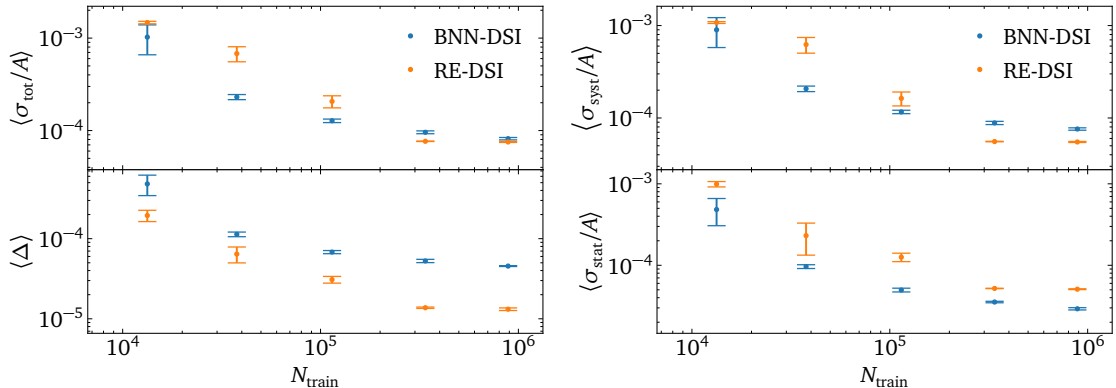

Figure 15: Left: relative total uncertainty for the BNN-DSI and RE-DSI, and relative accuracy. The means and error bars are obtained by averaging over five trainings. Right: relative uncertainties, split into systematics and statistics, as a function of the size of the training dataset.

**Sampled statistical uncertainties**

In Fig. 14, we show the sampled statistical pull, defined in Eq. (36) for the BNN-DSI and RE-DSI networks from our default training. In the left panel, we use only 10% of the training dataset, and both learned uncertainties are reasonably well calibrated given the limited amount of training data; in the right panel, we use the full training dataset. We see that the statistical uncertainty from the RE-DSI network is calibrated across small and large training datasets. In contrast, the statistical uncertainty of the BNN-DSI network becomes overconfident by roughly a factor of two for the full training dataset.

Finally, we look at the learned uncertainties of the DSI networks as a function of the size of the training dataset. In the left panel of Fig. 15, we compare the total relative uncertainties and the accuracies. The asymptotic values for the total uncertainty are similar for all training dataset sizes and coincide for the full dataset. The main difference between the two networks is that the accuracy of the RE-DSI is significantly better, because of the ensembling. We already know that the main reason for the mismatch between the accuracy and the uncertainty estimate is the poor calibration of the REs for the systematic uncertainties, where the learned uncertainty does not benefit from the ensembling the same way the learned amplitudes do.

The right panel of Fig. 15 splits the learned uncertainties into the relative systematic and statistical uncertainties for the BNN and RE. For small training datasets, the uncertainties learned by the two methods behave differently. This is expected for two reasons: First, the two methods approximate the posterior using different approaches with different implicit biases. Second, and more fundamentally, the separation of the total uncertainty into statistical and systematic contributions is not uniquely defined away from the limit of infinite statistics or negligible statistical uncertainties.

Towards more training data, the systematic uncertainties show a crossing point between BNN and RE, with the ensembles providing smaller uncertainties for large datasets. In this regime, the statistical uncertainties from the BNNs are slightly overconfident, while the statistical RE uncertainties are well calibrated, as we know from Fig. 14. On the other hand, the systematic uncertainties from the BNN are perfectly calibrated. This suggests that in splitting the total uncertainty into systematics and statistical parts, the BNN maintains a perfect calibration of the systematics through the heteroscedastic loss, at the expense of underestimating the statistical uncertainties by a factor of two.

## 7 Outlook

Fast, accurate, and controlled surrogate amplitudes are a key ingredient to higher-order event generation with future ML event generators. In terms of accuracy, standard MLP architectures have been surpassed, for instance, by a deep-set architecture with Lorentz invariants (DSI); an alternative path might be Lorentz-equivariant transformers (L-GATr).

We first studied the impact of activation functions on the accuracy using KANs and learnable activation functions through GroupKANs. While KANs perform worse than a well-chosen fixed activation function, GroupKANs yield comparable performance.

For deep networks, appropriate architectures can learn the uncertainties in parallel to the central values for amplitudes over phase space. Heteroscedastic losses in deterministic networks probe systematic uncertainties only, while BNNs and REs, combined with a heteroscedastic loss, track systematic and statistical uncertainties.

For systematic uncertainties, we found that a heteroscedastic loss and the BNN learn well-calibrated uncertainties. We tested this for added noise, network expressivity, and the symmetry implementation in the networks — in decreasing order of the size of the effect on the accuracy and the corresponding uncertainty. REs benefit from their ensemble nature in learning the mean amplitude but not in learning the systematic uncertainty. However, this significant mismatch could be removed through re-calibration. Importantly, we also showed that REs trained without a heteroscedastic loss do not learn any systematic uncertainties.

Statistical uncertainties are currently less relevant for LHC applications because networks are trained on comparably cheap simulations. However, for the DSI architecture, the BNN and the REs indicate that systematic uncertainties are reduced to the current level of statistical uncertainties. Calibrating the statistical uncertainties is conceptually challenging. We find that the BNN estimate is overconfident by roughly a factor of two, while the REs provide a calibrated statistical uncertainty.

Altogether, we have found that for surrogate loop amplitudes, learned uncertainties provide a meaningful way to control the training, identify challenges, and quantify the accuracy of the surrogates. They are key to understanding the improvement in relative accuracy from the percent level for naive networks to the $10^{-5}$ accuracy level for modern architectures. For the, in practice, most relevant systematic uncertainties, BNNs are sensitive to a wide range of sources of systematics and provide us with calibrated uncertainty estimates throughout.

Our implementation of KANs and Bayesian methods is available at https://github.com/heidelberg-hepml/arlo.

# Acknowledgments

**Funding information** RW is supported through funding from the European Union NextGeneration EU program - NRP Mission 4 Component 2 Investment 1.1 - MUR PRIN 2022 - Code 2022SNA23K. LF is supported by the Fonds de la Recherche Scientifique - FNRS under Grant No. 4.4503.16. NE is funded by the Heidelberg IMPRS *Precision Tests of Fundamental Symmetries*. This research is supported through the KISS consortium (05D2022) funded by the German Federal Ministry of Education and Research BMBF in the ErUM-Data action plan, by the Deutsche Forschungsgemeinschaft (DFG, German Research Foundation) under grant 396021762 – TRR 257: *Particle Physics Phenomenology after the Higgs Discovery*, and through Germany's Excellence Strategy EXC 2181/1 – 390900948 (the *Heidelberg STRUC-TURES Excellence Cluster*). Finally, we would like to thank the Baden-Württemberg Stiftung for financing through the program *Internationale Spitzenforschung*, project *Uncertainties - Teaching AI its Limits* (BWST_ISF2020-010).

# A    Additional material and hyperparameters

**Summary tables**

A detailed table for all networks versus noise split into different uncertainty contributions: In the last rows of Tab. 2, we show the total uncertainties learned by the two networks, by defining the square sum of the statistic and systematics. The BNN and RE results are similar. Just out of interest, we also show the uncertainty from the RE in a setup where we do not add the additional heteroscedastic loss. We see that it overestimates the total uncertainty without added noise and does not track the systematics from added noise at all. This indicates that the REs without heteroscedastic uncertainty are not well-suited to extract a systematic uncertainty like added noise.

Table 2: Learned uncertainties as a function of added noise, in terms of the median relative uncertainty for three amplitude quantiles. We use three hidden layers and show the learned systematic, statistical, and total uncertainties. For the latter, we also show the RE trained only with an MSE loss instead of a full heteroscedastic loss.

| | | 0% | 0.25% | 0.5% | 0.75% | 1% | 2% | 3% | 5% | 7% | 10% |
|---|---|---|---|---|---|---|---|---|---|---|---|
| $\langle\sigma_{\text{het}}/A\rangle$ | 25 | 0.0054 | 0.0061 | 0.0073 | 0.0094 | 0.012 | 0.021 | 0.030 | 0.051 | 0.072 | 0.104 |
| | 50 | 0.0047 | 0.0053 | 0.0068 | 0.0090 | 0.011 | 0.021 | 0.030 | 0.051 | 0.072 | 0.103 |
| | 75 | 0.0047 | 0.0053 | 0.0067 | 0.0089 | 0.011 | 0.021 | 0.030 | 0.051 | 0.073 | 0.104 |
| $\langle\sigma_{\text{syst, BNN}}/A\rangle$ | 25 | 0.0067 | 0.0069 | 0.0082 | 0.010 | 0.013 | 0.022 | 0.032 | 0.051 | 0.072 | 0.103 |
| | 50 | 0.0053 | 0.0060 | 0.0076 | 0.010 | 0.012 | 0.021 | 0.031 | 0.052 | 0.072 | 0.103 |
| | 75 | 0.0054 | 0.0059 | 0.0076 | 0.010 | 0.012 | 0.022 | 0.032 | 0.052 | 0.073 | 0.104 |
| $\langle\sigma_{\text{syst, RE}}/A\rangle$ | 25 | 0.0054 | 0.0061 | 0.0076 | 0.0095 | 0.012 | 0.021 | 0.031 | 0.050 | 0.070 | 0.101 |
| | 50 | 0.0045 | 0.0054 | 0.0070 | 0.0090 | 0.011 | 0.021 | 0.030 | 0.050 | 0.071 | 0.101 |
| | 75 | 0.0045 | 0.0052 | 0.0069 | 0.0090 | 0.011 | 0.021 | 0.030 | 0.050 | 0.070 | 0.101 |
| $\langle\sigma_{\text{stat, BNN}}/A\rangle$ | 25 | 0.0012 | 0.0013 | 0.0015 | 0.0018 | 0.0020 | 0.0028 | 0.0039 | 0.0061 | 0.0071 | 0.0084 |
| | 50 | 0.0012 | 0.0012 | 0.0014 | 0.0018 | 0.0019 | 0.0027 | 0.0037 | 0.0059 | 0.0072 | 0.0085 |
| | 75 | 0.0012 | 0.0013 | 0.0016 | 0.0019 | 0.0022 | 0.0031 | 0.0041 | 0.0066 | 0.0081 | 0.010 |
| $\langle\sigma_{\text{stat, RE}}/A\rangle$ | 25 | 0.0057 | 0.0059 | 0.0061 | 0.0065 | 0.0066 | 0.0076 | 0.0088 | 0.0106 | 0.012 | 0.015 |
| | 50 | 0.0046 | 0.0050 | 0.0053 | 0.0056 | 0.0057 | 0.0068 | 0.0078 | 0.0096 | 0.011 | 0.015 |
| | 75 | 0.0045 | 0.0048 | 0.0052 | 0.0055 | 0.0055 | 0.0066 | 0.0077 | 0.0094 | 0.011 | 0.013 |
| $\langle\sigma_{\text{tot, BNN}}/A\rangle$ | 25 | 0.0068 | 0.0071 | 0.0083 | 0.011 | 0.013 | 0.022 | 0.032 | 0.052 | 0.072 | 0.104 |
| | 50 | 0.0055 | 0.0061 | 0.0077 | 0.010 | 0.012 | 0.021 | 0.032 | 0.052 | 0.072 | 0.103 |
| | 75 | 0.0055 | 0.0061 | 0.0078 | 0.010 | 0.012 | 0.022 | 0.032 | 0.053 | 0.073 | 0.105 |
| $\langle\sigma_{\text{tot, RE}}/A\rangle$ | 25 | 0.0078 | 0.0085 | 0.0098 | 0.012 | 0.013 | 0.022 | 0.032 | 0.051 | 0.072 | 0.104 |
| | 50 | 0.0065 | 0.0073 | 0.0088 | 0.011 | 0.013 | 0.022 | 0.031 | 0.051 | 0.072 | 0.102 |
| | 75 | 0.0064 | 0.0071 | 0.0087 | 0.011 | 0.013 | 0.022 | 0.031 | 0.051 | 0.071 | 0.102 |
| $\langle\sigma_{\text{MSE, RE}}/A\rangle$ | 25 | 0.0081 | 0.0080 | 0.0082 | 0.0082 | 0.0084 | 0.0089 | 0.0096 | 0.0111 | 0.013 | 0.015 |
| | 50 | 0.0074 | 0.0073 | 0.0074 | 0.0075 | 0.0077 | 0.0081 | 0.0087 | 0.0101 | 0.011 | 0.014 |
| | 75 | 0.0073 | 0.0073 | 0.0073 | 0.0074 | 0.0075 | 0.0079 | 0.0085 | 0.0098 | 0.011 | 0.014 |

Table 3: Learned uncertainties as a function of added noise and the number of hidden layers, each with 128 dimensions. For the BNN with 4 or more hidden layers only the last layer is Bayesian, and the prior hyperparameter is $\sigma_{\text{prior}} = 0.316$ for 4 and 5 hidden layers and $\sigma_{\text{prior}} = 0.1$ for 6 hidden layers.

| | # layers | 0% | 0.25% | 0.5% | 0.75% | 1% | 2% | 3% | 5% | 7% | 10% |
|---|---|---|---|---|---|---|---|---|---|---|---|
| | 1 | 0.050 | 0.049 | 0.048 | 0.050 | 0.052 | 0.056 | 0.060 | 0.076 | 0.094 | 0.122 |
| | 2 | 0.010 | 0.011 | 0.012 | 0.014 | 0.015 | 0.023 | 0.033 | 0.052 | 0.071 | 0.102 |
| $\langle \sigma_{\text{Det}}/A \rangle$ | 3 | 0.0056 | 0.0062 | 0.0077 | 0.010 | 0.012 | 0.021 | 0.031 | 0.051 | 0.072 | 0.104 |
| | 4 | 0.0041 | 0.0048 | 0.0066 | 0.0086 | 0.011 | 0.020 | 0.031 | 0.052 | 0.073 | 0.105 |
| | 5 | 0.0038 | 0.0046 | 0.0061 | 0.0083 | 0.011 | 0.021 | 0.030 | 0.053 | 0.072 | 0.110 |
| | 6 | 0.0038 | 0.0043 | 0.0062 | 0.0085 | 0.012 | 0.021 | 0.031 | 0.054 | 0.072 | 0.102 |
| | 1 | 0.050 | 0.050 | 0.046 | 0.050 | 0.050 | 0.053 | 0.060 | 0.076 | 0.094 | 0.120 |
| $\langle \sigma_{\text{syst, BNN}}/A \rangle$ | 2 | 0.012 | 0.011 | 0.013 | 0.014 | 0.016 | 0.024 | 0.033 | 0.053 | 0.073 | 0.103 |
| | 3 | 0.0067 | 0.0071 | 0.0083 | 0.011 | 0.013 | 0.022 | 0.032 | 0.052 | 0.073 | 0.104 |
| | 4 | 0.0043 | 0.0049 | 0.0068 | 0.0090 | 0.011 | 0.021 | 0.033 | 0.051 | 0.073 | 0.103 |
| $\langle \sigma_{\text{syst, BNN}}/A \rangle$ | 5 | 0.0038 | 0.0068 | 0.0065 | 0.0091 | 0.014 | 0.023 | 0.033 | 0.054 | 0.073 | 0.103 |
| | 6 | 0.0034 | 0.0055 | 0.0063 | 0.0084 | 0.018 | 0.020 | 0.031 | 0.052 | 0.072 | 0.101 |

Table 4: Learned uncertainties as a function of added noise, in terms of the median relative uncertainty. We use the DSI network.

| | 0% | 0.25% | 0.5% | 0.75% | 1% | 2% |
|---|---|---|---|---|---|---|
| $\langle \sigma_{\text{syst, DSI RE}}/A \rangle$ | $5.1 \cdot 10^{-5}$ | 0.00249 | 0.00498 | 0.00753 | 0.0100 | 0.0205 |
| $\langle \sigma_{\text{syst, DSI BNN}}/A \rangle$ | $7.0 \cdot 10^{-5}$ | 0.00251 | 0.00499 | 0.00754 | 0.0100 | 0.0201 |
| $\langle \sigma_{\text{stat, DSI RE}}/A \rangle$ | $4.8 \cdot 10^{-5}$ | 0.00014 | 0.00025 | 0.00042 | 0.00068 | 0.00136 |
| $\langle \sigma_{\text{stat, DSI BNN}}/A \rangle$ | $2.3 \cdot 10^{-5}$ | 0.00016 | 0.00026 | 0.00070 | 0.00083 | 0.0014 |
| $\langle \sigma_{\text{tot, DSI RE}}/A \rangle$ | $7.0 \cdot 10^{-5}$ | 0.00250 | 0.00500 | 0.00756 | 0.0100 | 0.0205 |
| $\langle \sigma_{\text{tot, DSI BNN}}/A \rangle$ | $7.4 \cdot 10^{-5}$ | 0.00451 | 0.00506 | 0.00757 | 0.0101 | 0.0201 |

**Hyperparameters**

Table 5: Network and training parameters of the MLP/DSI.

| Parameter | MLP | DS(I) |
|---|---|---|
| Size of latent rep. | - | 64 |
| Activation function | ReLU | GELU |
| Number of layers | 3 | 3 |
| Hidden nodes | 128 | 128 |
| Batch size | 1024 | 1024 |
| Scheduler | One cycle | Cosine |
| Max learning rate | $10^{-3}$ | $10^{-3}$ |
| Number of epochs | 1000 | 1000 |

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
