# Peer review of "Accurate Surrogate Amplitudes with Calibrated Uncertainties"

_SciPost Physics, doi:SciPost Phys. Core 8, 073 (2025)_

## Round 1 · Referee Report · Anonymous (Referee 1) · 2025-5-27

Strengths

This work advances the toolbox of machine learning in HEP by developing methods for predicting uncertainties together with the target value of a (as an example) amplitude function.

The mathematical derivations are precise and pedagogical, quite in contrast to much of the ML literature in the field, and will benefit a wider audience.

Because HEP is unique in the depth of mathematical modelling, new non-HEP ML techniques rarely satisfy requirements on uncertainty quantification. This work is an exception as it fills several gaps in this regard, and opens a door for making future algorithms uncertainty-aware.

The studies are comprehensive and probably generalize much beyond the considered examples.

Weaknesses

The emphasis of the text is clearly on the maths.

The language of the main body of the text should be improved, and often does not sufficiently carry the reader through the rather intricate reasoning. This can be seen implicitly in the various small comments I attach below and the general remarks.

A simple way to phrase my main comment would be that the current text requires the reader to have already digested the ideas and the main developments of the paper. This can not be expected. The text should address the reader at a much earlier level by providing relevant context and guidance at each step. This does NOT mean to inflate the jargon (of which there is enough) or merely the word count (which is good), but to use precise and lean language that carries meaning in a more condensed, logical, and structured way.

Report

Acceptance criteria are clearly met. The major revision pertains to the Text.

Requested changes

General remarks:

The reader is not sufficiently prepared in the abstract and introductory sections to follow the reasoning. Because I believe a paper should be comprehensible on the first pass, I advise the author to carefully re-read the manuscript and, in each paragraph, ask whether it is clear to the reader where we are going, what to expect, and what the next steps are. In particular, there should be carefully introductory paragraphs that explain in words what the developments are.

The long derivations are a strength, but I believe it is distracting to the narrative. The general derivations should go to an appendix, and the main body of the text should discuss the specifics, adopting a precise language free from jargon. For example, most of the derivations of Sec. 2.2, 2.3, 2.4, and 4 are not necessary to understand the results.

As an exemplary comment:
Types of uncertainties should be defined more precisely. There is a section on p4, but the reader is not prepared for the (in hindsight) very general derivation. On the other hand, it is left unclear what the exact relation is to the statistical and experimental systematic uncertainty in HEP language, as well as what the ML domain refers to as aleatoric and epistemic. While the derivation is not too long, I do not think that so much is gained from the rule of total uncertainty. Instead, the reader should be able to understand from the text how each component is understood.

Detailed comments:

Abstract:

(General) I believe the abstract should describe what the goal of the paper is, what tools are used, and what the result is.
Currently, the sentences are not logically connected unless the reader has already read the paper.

L3 KANs were not introduced.
L5 comprehensive"ly"
"precision surrogates" is unclear

p2
Paragraph 3.
"amplifying" "essentially interpretable" are jargon and unclear
Paragraph 5. Where does the percent-level requirement come from? This is not true in general and highly application specific.
Paragraph 6. "tested assumption" what does this mean?

p5
After Eq. 13 sentence not finished.
p7 "In reality" is too colloquial and does not add information
p8 "Realistically, ..." same.
p9 "define the underlying problem." A problem is set or solved, but not defined.

Sec 4. The choice for KANs is not sufficiently explained, here or in the introduction.

p13, one but last paragraph. This is a typical example of my comments. It is not clear to the reader or explained (as far as I see) why a flatter activation function is desirable. This leaves the reader with the impression they are not on the same page as the author.

Fig. 4. The imperfections at low f_smear are interesting. Please add general remarks on implications or limitations in general. Clearly, this is a hint of a ceiling for any such methods and of general interest.

p18 "Bayesianize" is entirely unclear

Recommendation

Ask for major revision

---

## Round 1 · Referee Report · Anonymous (Referee 2) · 2025-6-9

Report

In this manuscript the authors provide a detailed study how well neural networks can approximate
scattering amplitudes. As an example, the loop-induced scattering amplitude $g g \rightarrow \gamma \gamma g$ is considered.
The authors distinguish statistical uncertainties and systematic uncertainties.
By definition, statistical uncertainties vanish in the limit of infinite training data, systematic uncertainties are related
for example to the network architecture.
The major part of the manuscript is very well written: Section 2 gives a very good introduction to learned uncertainties,
section 3 defines the concrete scattering amplitude and the neural networks considered, section 5 and 6 present the results
for the systematic and statistical uncertainties, respectively.
The only section, which appears rather unmotivated and breaks the flow of reading, is section 4 on KAN amplitudes.
It consists of two parts, one introductory parts for Kolmogorov-Arnold networks and a second part on the study of activation
functions.
The manuscript could profit if the authors include between section 2 and 3 a further section (in the style of section 2),
where they briefly introduce multi-layer perceptron networks, deep sets networks, geometric algebra transformer networks
and the concept of activation functions.
The introductory parts for Kolmogorov-Arnold networks can also be part of this section and the authors could motivate a little
bit more why they would like to study the difference between fixed activation functions and learned activation functions.
Afterwards they could continue with the previous section 3 "Amplitude data and network architectures" and the remaining
part "Activation functions" of the previous section 4.
This would have the advantage that it clearly divides the manuscript on the one hand into a general introductory part,
which does not depend on scattering amplitudes, and on the other hand the concrete study of the $g g \rightarrow \gamma \gamma g$
scattering amplitude.

Minor comments:

Page 2, second paragraph: one reference appears as "?".

Section 3: The full name for the abbreviations MLP and GATr can be given as well.

Section 4: ReLU, GELU, leakyReLU: these are standard functions for machine learning, but for readers from particle physics
the full name will be helpful. The authors might also consider providing the definitions in an appendix.

Requested changes

See above.

Recommendation

Ask for minor revision

---

## Round 2 · Referee Report · Anonymous (Referee 2) · 2025-9-9

Report

The two referee reports from the first round gave the authors indications
on how to improve the manuscript for a wider audience.
The authors didn't really pick up on that.
In the revised version the authors try to limit the modifications
as far as possible to a minimum.
The scientific content is worth a publication, but as the authors give the
impression that they are only interested in addressing readers in their
particular field of research,
SciPost Physics Core is maybe the better place for this manuscript.
I leave this decision to the editor.

Recommendation

Accept in alternative Journal (see Report)

---

## Round 2 · Referee Report · Anonymous (Referee 1) · 2025-9-9

Report

The authors have implemented most suggestions, and the text is significantly improved.

Recommendation

Publish (surpasses expectations and criteria for this Journal; among top 10%)

---

## Round 2 · Author Response

Dear Editor and Referees,

We thank the referees for their time, careful consideration, and positive evaluation of our manuscript. We list below the changes we have made concerning the helpful suggestions.

General remarks:

We conclude the KAN section, showing that the learned activation function gives us useful information to decide the best activation function for the standard neural networks. Fig.4 shows that the GELU activation function, approximately represented by a learned GroupKAN activation function, is preferred over the standard ReLU.

We share our implementation of KANs and Bayesian networks with the community. Our code Detailed comments:is available at https://github.com/heidelberg-hepml/arlo .

We hope that with these changes, our article can now be accepted for publication in its present form.

Sincerely,

H. Bahl, N. Elmer, L. Favaro, M. Haußmann, T. Plehn, R. Winterhalder

---

## Round 2 · List of Changes

Referee 1:

We state the relation to uncertainties as defined in the ML domain at the end of Section 2. As the ML definition is often vague and should be characterised on a case-by-case basis, we prefer to use an operational definition based on behaviour in the large training statistics regime. We prefer to keep the derivations of Sections 2 and 4 in the main body of the paper. We believe that they allow the reader to understand the logical progression from the theoretical Bayesian theory to the practical implementation of learned uncertainties. The full derivation also ensures that all the assumptions and approximations are discussed, allowing future readers to have an instructive reference which can be used to build upon this work.

Abstract: (General) I believe the abstract should describe what the goal of the paper is, what tools are used, and what the result is. Currently, the sentences are not logically connected unless the reader has already read the paper.

We made the abstract more logically coherent.

L3 KANs were not introduced.

When introducing the KANs in the introduction, we added additional motivation for studying them (a promise for better accuracy and scalability). This is mentioned again at the beginning of Sec. 4.

L5 comprehensive"ly" "precision surrogates" is unclear p2 Paragraph 3. "amplifying" "essentially interpretable" are jargon and unclear Paragraph 5. Where does the percent-level requirement come from? This is not true in general and highly application specific. Paragraph 6. "tested assumption" what does this mean?

We updated the introduction, taking into account these comments. In particular, we clarified the “amplification” argument. We agree that the percent-level requirement is application-specific and therefore removed the sentence. We rephrased the end of paragraph 6 to clarify our expectations for the statistical uncertainties.

p5 After Eq. 13 sentence not finished.

Fixed.

p7 "In reality" is too colloquial and does not add information p8 "Realistically, ..." same. p9 "define the underlying problem." A problem is set or solved, but not defined.

We removed several colloquial terms from the paper, including the ones above, and removed the unnecessary sentence in the scaled pull section.

Sec 4. The choice for KANs is not sufficiently explained, here or in the introduction.

When introducing the KANs, we added additional motivation for studying them (a promise for better accuracy and scalability). This is mentioned again at the beginning of Sec. 4.

p13, one but last paragraph. This is a typical example of my comments. It is not clear to the reader or explained (as far as I see) why a flatter activation function is desirable. This leaves the reader with the impression they are not on the same page as the author.

Flatter activation functions are not the scope of the study. In the final paragraphs of the KAN section, we discuss the general behaviour of the learned activation functions, and we observe that they can be used as a baseline for finding the best option among the fixed activations of a standard neural network. See general remarks.

Fig. 4. The imperfections at low f_smear are interesting. Please add general remarks on implications or limitations in general. Clearly, this is a hint of a ceiling for any such methods and of general interest.

The imperfections for a small smearing applied to the data originate from the intrinsic uncertainties of the network architectures. There is no infinitely expressive architecture which learns the data distribution exactly. This residual systematic effect can be seen as the deviation from the ``noise only” curve. We added this discussion to Section 5.1.

p18 "Bayesianize" is entirely unclear

We agree with the referee and we now specify that only a subset of the weights have learnable variance while the rest of the neural network is deterministic.

We revised the manuscript following the referee’s suggestion and made these additional changes:

  • At the beginning of Section 2, we clarify that the space $x$ is the space of four-momenta of the scattering particles.
  • In the same section, we state that the derivations of the various loss functions generalise beyond amplitude regression.
  • On pg.4, we clarify that a Gaussian ansatz for the posterior distribution does not imply a Gaussian posterior distribution.
  • We removed the derivation of Eq.8.
  • In Section 2.3, we clarified the role of $t$ as the iteration during training, and we discuss in more detail the pitfalls of a simple ensemble of networks.
  • In the “Function-space density” section, we rephrased the motivation for introducing a function space repulsion term over the weight-space one.

Referee 2:

We believe that multi-layer perceptron networks and activation functions are well-known and have been widely discussed in the field. We think an introduction is unnecessary, and it would rather dilute the novel methodologies we are presenting in the manuscript. We reformulated the introduction of section 3 (and also the text on KANs in the introduction), making it logically more connected to the rest of the text.

Minor comments:

Page 2, second paragraph: one reference appears as "?".

Fixed.

Section 3: The full name for the abbreviations MLP and GATr can be given as well.

The full names are now given.

Section 4: ReLU, GELU, leakyReLU: these are standard functions for machine learning, but for readers from particle physics the full name will be helpful. The authors might also consider providing the definitions in an appendix.

We now specify that ReLU, GELU, and leakyReLU belong to the rectified linear unit family of fixed activation functions.

---

## Editorial Decision

published